# Smoke-charged vortices in the stratosphere generated by wildfires and their behaviour in both hemispheres : comparing Australia 2020 to Canada 2017

Hugo Lestrelin, Bernard Legras, Aurélien Podglajen, and Mikail Salihoglu

Laboratoire de Météorologie Dynamique, UMR CNRS 8539, IPSL, PSL-ENS/École Polytechnique/Sorbonne Université, Paris, France

**Correspondence:** Bernard Legras (bernard.legras@lmd.ipsl.fr)

**Abstract.** The two most intense wildfires of the last decade that took place in Canada in 2017 and Australia in 2019-2020 were followed by large injections of smoke in the stratosphere due to pyro-convection. It was discovered by Khaykin et al. (2020) and Kablick et al. (2020) that, after the Australian event, part of this smoke self-organized as anticyclonic confined vortices that rose in the mid-latitude stratosphere up to 35 km. Based on CALIOP observations and the ERA5 reanalysis, this new study analyzes the Canadian case and finds, similarly, that a large plume penetrated the stratosphere by 12-13 August 2017 and got trapped within a meso-scale anticyclonic structure which travelled across the Atlantic. It then broke into three offsprings that could be followed until mid-October performing three round the world journeys and rising up to 23 km. We analyze the dynamical structure of the vortices produced by these two wildfires and demonstrate how the assimilation of the real temperature and ozone signatures data from instruments measuring the signature of the vortices explains the appearance and maintenance of the vortices in the constructed dynamical fields. We propose that these vortices can be seen as bubbles of small, almost vanishing, potential vorticity and smoke carried vertically across the stratification from the troposphere inside the middle stratosphere by their internal heating, against the descending flux of the Brewer-Dobson circulation.

## 1 Introduction

A spectacular consequence of large summer wildfires in mid-latitude forests is the generation of pyro-cumulonimbus (PyroCb) that can reach the lower stratosphere during extreme events (Fromm et al., 2010). The combustion products and accompanying tropospheric compounds (organic and black carbon, smoke aerosols, condensed water, carbon monoxide, low ozone, ...) that are lifted to the stratosphere can survive several months and be transported over considerable distances (Fromm et al., 2010; Yu et al., 2019; Kloss et al., 2019; Bourassa et al., 2019) filling the mid-latitude band and sometimes reaching the tropics (Kloss et al., 2019). As black carbon is highly absorptive of the incoming solar radiation, the resulting heating produces buoyancy (de Laat et al., 2012; Ditas et al., 2018) and an additional lift of the non diluted parts of the plume by several kilometers in the stratosphere (Khaykin et al., 2018; Yu et al., 2019). This effect enhances the dispersion and, by increasing the altitude, ensures a longer life-time in the stratosphere (Yu et al., 2019), enhancing the radiative effect on climate that has been estimated of comparable amplitude to moderate volcanic eruptions (Peterson et al., 2018; Khaykin et al., 2020).

The Australian wildfires of the summer season 2019-2020 have exceeded in magnitude all previously known events (Khaykin et al., 2020). A striking discovery has been the observation that a part of the stratospheric smoke plumes self-organized as anticyclonic vortices that persisted between one to three months (Kablick et al., 2020; Khaykin et al., 2020; Allen et al., 2020) and, for the most intense one (nicknamed below as Koobor, following an aboriginal legend), rose up to $35\,\mathrm{km}$ (Khaykin et al., 2020), an altitude never reached by tropospheric aerosols since the Pinatubo eruption. It was conjectured by Khaykin et al. (2020) that aerosol heating was essential in maintaining the structure and providing the lift. In turn, the vortex created a confinement that preserved the embedded smoke cloud from being rapidly diluted within the environment.

Investigating the occurrence of such vortices after previous wildfires, in particular over the last 15 years during which the required satellite instruments are available, is a natural extension of (Khaykin et al., 2020). As the strongest previously recorded wildfire of the last decade is the 2017 Canadian event that took place in British Columbia (Hanes et al., 2019), this work is revisiting this case which has already been documented in several studies (Khaykin et al., 2018; Ansmann et al., 2018; Peterson et al., 2018; Yu et al., 2019; Kloss et al., 2019; Baars et al., 2019; Torres et al., 2020). In particular, a stratospheric rise of up to $30\,\mathrm{K\,day^{-1}}$ in potential temperature was diagnosed based on satellite observation by Khaykin et al. (2018) who also report a compact smoke cloud at $19\,\mathrm{km}$ over the Haute-Provence observatory, Southern France, on 29 August 2017. An other goal of this study is to complement Khaykin et al. (2020) by expanding their diagnostics and interpretations on the 2020 case.

Section 2 describes the data and methods used in this work. Section 3 describes the new vortices found after the 2017 Canadian fire and their evolution. It includes a detailed discussion of previous results. Section 4 describes the structure of the vortices based on the 2017 Canadian case and the 2020 Australian case. Section 5 offers conclusions.

## 2 Data and methods

### 2.1 Satellite data from CALIOP

Launched in April 2006, the Cloud-Aerosol Lidar and Infrared Path Satellite Observation (CALIPSO) mission (Winker et al., 2010) took on board the Cloud-Aerosol Lidar with Orthogonal Polarization (CALIOP) which is a two-wavelength polarization lidar that performs global profiling of aerosols and clouds in the troposphere and lower stratosphere. We used the total attenuated $532\,\mathrm{nm}$ backscatter level 1 (L1) product in the latest available version V4.10 (Powell et al., 2009). The nominal along track horizontal and vertical resolutions are, respectively, $1\,\mathrm{km}\,/\,60\,\mathrm{m}$ between 8.5 and $20.1\,\mathrm{km}$ and $1.667\,\mathrm{km}\,/\,180\,\mathrm{m}$ between 20.1 and $30.1\,\mathrm{km}$. The L1 product oversamples the layers above $8.5\,\mathrm{km}$ with an uniform horizontal resolution of $333\,\mathrm{m}$. We computed the scattering ratio by dividing the total attenuated backscatter by the calculated molecular backscatter, following Vernier et al. (2009) and Hostetler et al. (2006), and using the meteorological metadata of the L1 product. To reduce the noise, an horizontal median filter with $40\,\mathrm{km}$ width (121 pixels) was applied to the data. In order to separate clouds from aerosols, we also used the level 2 (L2) total scattering aerosol coefficient at $532\,\mathrm{nm}$ which is available at $5\,\mathrm{km}$ resolution. Both daytime and nighttime measurements have been used. Since the daytime measurements are noisier, they can only be used when the aerosol signal is strong enough, mainly during the first weeks after the release of the plume.

As in Khaykin et al. (2020), CALIOP inspection has been the first step in order to identify the potential vortices. The L1 sections have been systematically screened from 12 August to 15 October 2017, selecting those which contained isolated compact patterns above 11 km, identified as aerosols by the L2 product. The location and size of the retained patches were then determined by visually matching a rectangular box to the observation, as illustrated in Fig. A1. It is usually very easy to see the boundaries of the retained patches. At the next stage, the patches have been associated to the vortices detected as described below, and a further inspection has rejected cases that corresponds to tails left behind by vortices (less than 20% of those retained at the first stage).

Due to an increased solar activity, CALIOP operations have been suspended between 5 and 14 September. This has been an obstacle to establish a continuity based on CALIOP observations alone but the tracking of the vortices filled that gap as described below.

## 2.2 Meteorological data

### 2.2.1 Reanalysis

To track the stratospheric wildfire vortices and diagnose their dynamical structure, we used the ERA5 reanalysis (Hersbach et al., 2020) which is the last generation global atmospheric reanalysis of the European Center for Medium Range Weather Forecast (ECMWF). Khaykin et al. (2020) used instead the ECMWF operational analysis and forecast. Both are based on the ECMWF Integrated Forecast System (IFS). In the ERA5, the native horizontal resolution of the IFS model is about 31 km and it has 137 levels in the vertical with spacing that varies from less than 400 m at 15 km to about 900 m at 35 km within the relevant altitude range of this study. We used an extracted version of the ERA5 with 1°resolution in latitude and longitude, at full vertical resolution and 3-hourly resolution. This choice was dictated by practical considerations, and by the evidence that this grid is able to describe synoptic-scale features and that vortices do not travel across more than a few grid points in 3 hours.

Khaykin et al. (2020) used temperature, ozone and vorticity to investigate the structure of the vortices. In this study, we also used potential vorticity (PV). PV is a Lagrangian invariant for inviscid and adiabatic flows (Ertel, 1942); furthermore, it provides a compact and complete picture of the balanced part of the flow (Hoskins et al., 1985). An interesting property of the ERA5 is that its dynamical core preserves PV much better than previous reanalyses (Hoffmann et al., 2019). Although ERA5 potential vorticity can be directly retrieved from the ECMWF archive on a given set of potential temperature levels, we instead recalculated it from the retrieved vorticity, temperature and total horizontal wind fields on model levels, in order to benefit from the full vertical resolution offered by ERA5. For that purpose, we used the definition of PV for the primitive equations in spherical coordinates and model hybrid vertical coordinate $\eta$, which is given by :

$$P = -g \left( \frac{\partial \theta}{\partial p} (f + \zeta_\eta) + \frac{1}{a} \left( \frac{\partial \theta}{\partial \phi} \bigg|_\eta \frac{\partial u}{\partial p} - \frac{1}{\cos \phi} \frac{\partial \theta}{\partial \lambda} \bigg|_\eta \frac{\partial v}{\partial p} \right) \right) \tag{1}$$

where $a$ is the radius of the Earth, $\lambda$ is the longitude, $\phi$ the latitude, $g$ is the free-fall acceleration, $\zeta_\eta$ is the model level vertical component of the vorticity, $f$ the Coriolis parameter, $(u, v)$ the horizontal velocity, $p$ the pressure and $\theta$ the potential

temperature. The gradients are estimated using centered differences on the retrieved longitude-latitude-$\eta$ grid. See Eqs. (3.1.4) of Andrews et al. (1987) for an analogous formula in barometric altitude coordinate.

While the formulation of PV in Eq. (1) is the most commonly used, it bears the disadvantage of a large background vertical gradient, which proves inconvenient to track and characterize structures along their ascent. To overcome this issue, we used the alternative formulation of Lait (1994), discussed by Müller and Günther (2003):

$$\Pi = P \left( \frac{\theta}{\theta_0} \right)^{-\epsilon} \tag{2}$$

where $P$ is the Ertel PV, $\epsilon = 4$ in the Australian case and $\epsilon = \frac{9}{2}$ in the Canadian case, and $\theta_0 = 420\,\mathrm{K}$. Compared to $P$, $\Pi$ is still an adiabatic invariant and exhibits a reduced vertical gradient, so that the vortices, characterized by anticyclonic $\Pi$ anomalies, can be unambiguously distinguished during their ascent. In each of the two cases, the value of $\epsilon$ is chosen to nearly cancel the background vertical gradient of $\Pi$, and depends on the large-scale vertical temperature profile characterized by larger (more positive) $\frac{\partial T}{\partial z}$ in the Australian case than in Canadian case (see Müller and Günther, 2003, for a discussion regarding the choice of $\epsilon$).

### 2.2.2 Assimilation increment

The ERA5 is constrained by observations over repeated 12-hour assimilation cycles. Over each cycle, the assimilation increment is defined as the difference between the new analysis and the first guess provided as a final stage of a free forecast run of the model, initialized from the previous analysis 12 hours before. This definition can be applied to any of the basic variables of the model or to derived quantities like potential vorticity. In the ERA5, the assimilation increments can be calculated on each day at 6:00 and 18:00 UTC. In order to diagnose how the observations are forcing the vortices, we calculated the assimilation increments of temperature, vorticity, potential vorticity and ozone. Temperature and ozone determine the radiances that are measured by spaceborne instruments and are also directly accessible from in situ instruments. On the contrary, potential vorticity cannot be directly retrieved from any instrument and is indirectly constrained (see below). These three parameters are updated by the assimilation system in order to reduce the difference between observed quantities (typically radiances but also deviations of the GPS signal path) and simulated quantities (radiances that a satellite flying "above the model" would see). It is tempting to see the temperature assimilation increment as an additional heating but this is incorrect. The increment is calculated from an adjusted state, resulting from the iterations of the assimilation, in which both temperature and motion respond to the forcing by the observations. As wind observations are much sparser than temperature observations, one would expect that analysis winds, and related quantities like potential vorticity, are more poorly constrained and therefore less accurate than analysis temperatures. While this statement is true to a large extent in the tropics, in the mid-latitudes the temperature and wind fields are related through thermal wind balance. This equilibrium is enforced by the assimilation system which filters out the transient modes that deviate from it. Hence, thanks to this *mi*racle of assimilation, assimilating the temperature signal of the vortex is sufficient to reconstruct the whole thermal and dynamical field associated with the balanced structure (McIntyre, 2015).

It should be noted here that neither the ECMWF operational analysis nor the ERA5 assimilate aerosol observations. The smoke plumes are totally absent from the IFS, where stratospheric aerosols were only accounted by mean climatological distribution during the periods of investigation, and it is only their dynamical vortical signature which are introduced in the model as described above.

### 2.2.3 Vortex tracking approach

Once we caught a first occurrence of an isolated bubble of aerosols with CALIOP, we searched the ERA5 data for the occurrence of a corresponding vortex that showed up as an isolated pattern in the $\Pi$ and ozone maps. While Khaykin et al. (2020) used only relative vorticity to track the vortices for commodity, we used both $\Pi$ defined from Eq. (2) and the ozone anomaly defined as the deviation with respect to the zonal mean at the same latitude and altitude. The tracking was made every six hours by following a local extremum within a box of usually 12 degrees in longitude, 5 degrees in latitude and a range of at least 30 K of potential temperature in the vertical. Once a vortex was caught, the box was moved forward in time to the step $n+1$ according to the vortex motion between steps $n-1$ and $n$. In very few instances, the tracking was guided by reducing the size of the box. This was needed in particular at the formation of a vortex, or when it split in two parts and in the final stage when one of the methods lost the track before the other.

## 3 A new occurrence : Canada 2017

### 3.1 General description

Although the 2017 Canadian fire started in June (Hanes et al., 2019), it is only by 12 August until early 13 August that the fire reached an intensity such that a large PyroCb developed and reached the lower stratosphere leaving a smoke plume that could be followed by satellite sensors (Khaykin et al., 2018; Peterson et al., 2018; Torres et al., 2020). From the inspection of scattering ratio along CALIPSO orbits, we found our first distinctly isolated smoke bubble on 16 August at 9:38 UTC and 12.6 km altitude, over the north of Canada at $63°$ N and $102°$ W, where the tropopause altitude was 11 km. This is two days after the first evidence of presence of smoke in the stratophere (Peterson et al., 2018; Torres et al., 2020). In the following days and until the CALIOP interruption on 4 September, we could track this cloud, labelled as bubble O, and its offsprings almost every day. Figure 1 shows several typical patterns during that period.

As indicated by the longitudes on Fig. 1, the bubble O moved eastward across the Atlantic. During the early days, it was captured each day by at most two CALIPSO orbits (one day orbit and one night orbit), the adjacent orbits only showing filamentary non compact structures that could be easily distinguished. The bubble rose rapidly reaching 18 km by 25 August (i.e. an ascent rate larger than $0.5 \, \mathrm{km \, day^{-1}}$). One surprising feature is that once it reached the European coast by 27 August, several bubbles could be tracked on multiple close orbits, or even exceptionally, on the same orbit. As we shall see in the following, this corresponds to the splitting of bubble O into several offsprings that we label as A, B1 and B2 for which several views are shown in Fig. 1. Another issue came soon after as CALIOP operations were suspended for the period 5 - 14 September

Aerosol scattering ratio 512

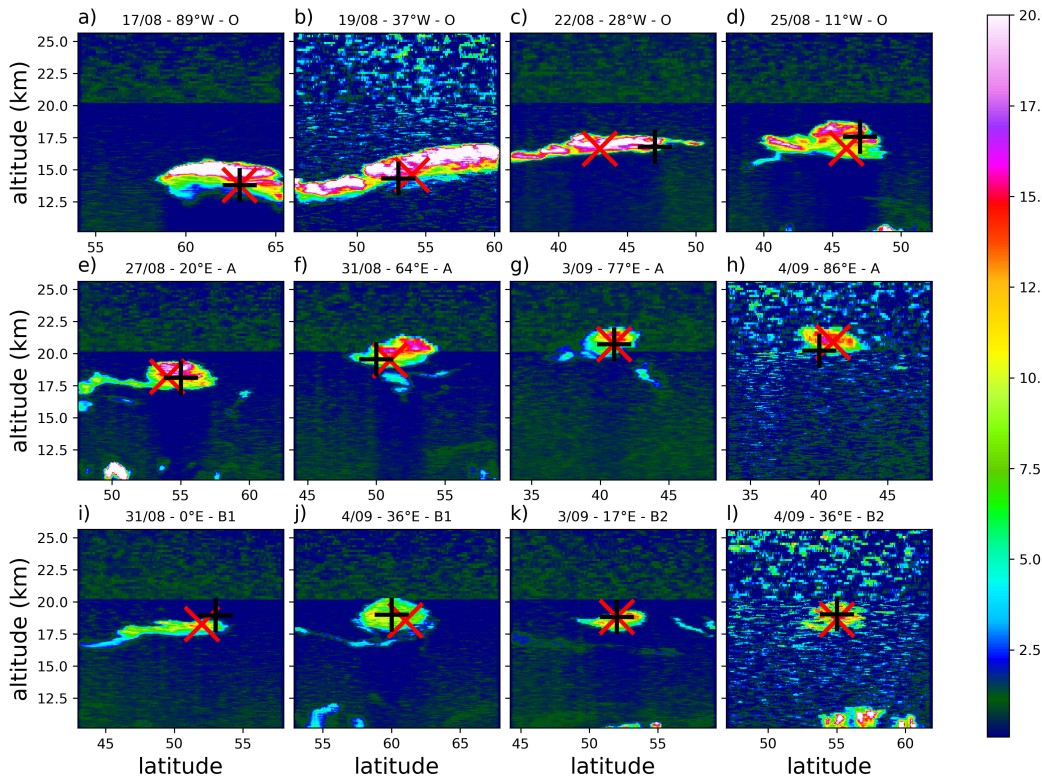

**Figure 1.** Selection of along track sections of CALIOP scattering ratio profiles during the first observation period until 4 September 2017. The upper row (a-d) shows sections of the bubble O at four times. The middle row (e-h) shows four sections of the bubble A after its separation from bubble O. The bottom row shows on the left (i-j) two sections of bubble B1 after its separation from bubble O and on the right (k-l) two views of bubble B2 that continues the track of bubble O after 1st September 2017. In each panel, the black and red crosses show the orbital plane projection of the corresponding vortex center according to, respectively, the Lait PV Π and the ozone trackings in the ERA5 data. On each panel, the longitude indicated in the title is that of the CALIPSO orbit at the center of the bubble.

due to increased solar activity. When CALIOP observations resumed after 14 September, the A, B1 and B2 bubbles that were now located at 20 km or above could be found again, over central Asia for A and over the Pacific for B1 and B2, and they were followed during their subsequent journey until mid October. Figure 2 shows a selection of views during that period.

## 3.2 Early evolution

The smoke bubbles have been attributed to vortices based on the ERA5 reanalysis which were available during their whole life cycles. Starting on 14 August, a kernel of almost zero PV and low ozone could be found by 72° N and 115° W at an altitude of 12 km just above the tropopause. In the following days, this anomaly developed vertically and connected with the

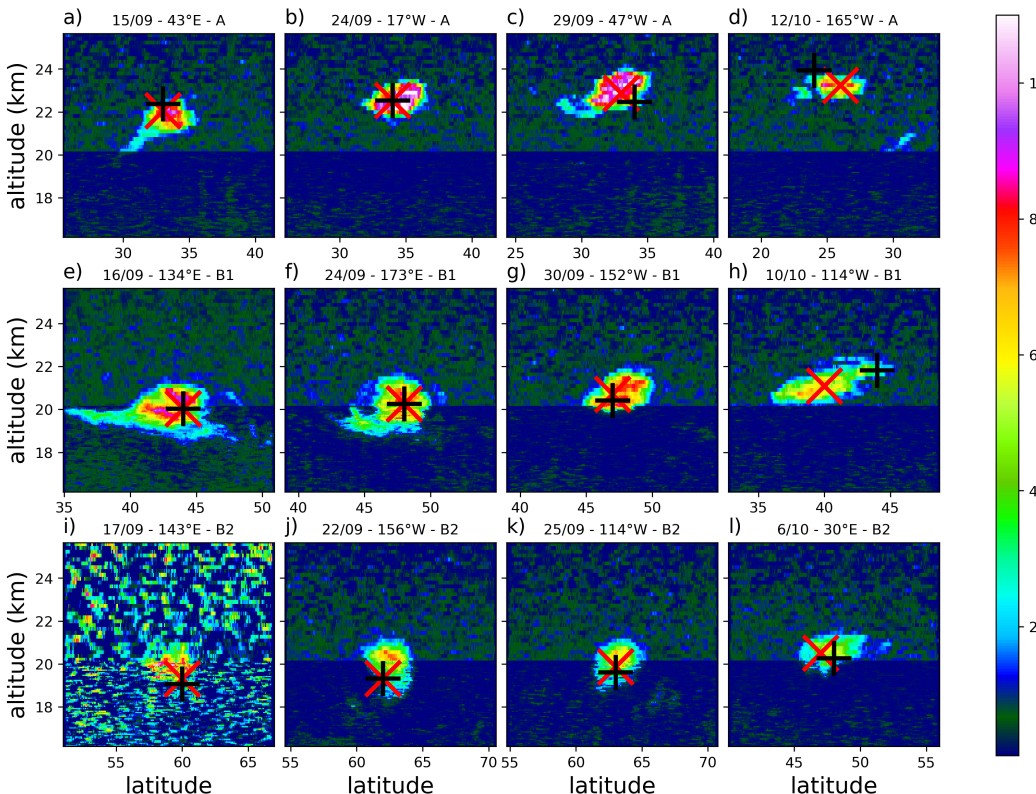

**Figure 2.** Same as Fig. 1 but during the second observation period after 15 September 2017 and showing, respectively, in the upper, middle and bottom rows, four sections of bubbles A (a-d), B1(e-h) and B2 (i-l).

bubble O location as identified from CALIOP (see animation in Sec. S2 of the Supplement). By 17 August, as it crossed the Hudson Bay, it exhibited a well developed intrusion that reached 14 km in the PV longitudinal and latitudinal section. As seen in the animation, this intrusion, while still rising, was subsequently stretched by the vertical shear and split by 19 August into an upper part isolated in the stratosphere at about 15 km while the lower part near the tropopause was further stretched and disappeared. The upper part which was associated with bubble O was tracked in ERA5 ozone and PV fields and from CALIOP until the end of August (Fig. S1 of the Supplement). As it crossed the Atlantic, it got trapped inside a trough by 20 August and travelled with it until it reached the European coast. Due to the wind shear prevailing in the associated jet, the vortex O elongated in latitude across the isentropes (see the animation) until it got split over western Europe into the three parts A, B1, B2, as described in the next section.

This early stage is also described in great detail by Torres et al. (2020). These authors show that the whole smoke cloud is shaped, on 20 August, as a V by its passage in the trough resulting in a double maximum tilted pattern in the section by

CALIOP. This pattern is recovered in the ERA5 PV pattern (freeze the animation in Sec. S2 of the Supplement as the same date) and our tracking is actually following the highest of the two maxima.

## 3.3 Horizontal and vertical splitting of vortex O into its offsprings

Figure 3 displays the series of events that led, over the period 22 August - 1st September, to the splitting of the vortex O into its three offsprings that were subsequently followed during one month and a half. We display the CALIPSO orbits that intersect the vortex cores and also those that intersect their tails. The presence of a smoke bubble or patch detected by CALIOP along an orbit is shown as a red segment. We see that on all orbits, there are bubbles or patches that match each of the intersections with the tracked low PV regions. The sequence begins, on 22 August (Fig. 3a) with the elongated structure of the vortex O that emerged at 420 K on the eastern flank of the trough within which it crossed the Atlantic. The elongation is due to the intense vertical shear in the jet stream. The formation of vortex A is already visible as the wrapping up pattern on the north east end of vortex O. On 23 August (Fig. 3b), the pattern of vortex A is lost at level 420 K but is now visible at level 435 K. Two days later (Fig. 3c), the vortex A was a developed structure reaching $60°$ N at level 455 K which was overflown by CALIPSO (Fig. 4a). At the same time the vortex O maintained a core near $45°$ N (Fig. 1d). In the following days, on 26 and 27 August (Fig. 3d-e), the vortex A fully separated from vortex O as it moved eastward and rose (Fig. 1e). On 28 and 29 August (Fig. 3f-g), the vortex A moved away while the vortex O was anew elongated, and started to split into a western part (Fig. 4b) and an eastern part (Fig. 4c). The vertical structure of vortex A on 28 August in an other reanalysis is described in Fig. 16 of Allen et al. (2020). On 30 and 31 August (Fig. 3h-i), the western part separated while the eastern part folded itself and separated in two more parts that provided vortex B1 as the north component (Fig. 1) and vortex B2 as the south component (essentially the continuation of the vortex O) which were both seen on the same CALIOP overpass of 1st September (Figs. 3j & 4d). The vortices B1 and B2 fully separated in the following days while rising and starting to move slowly eastward. The western component could also be followed until September 4, accompanied by patches seen by CALIOP. It remained, however, below 460 K and could not be linked with certainty to any structure seen from CALIOP after 15 September.

## 3.4 Late evolution

During the observations period that followed the recovery of CALIOP after 15 September and until mid-October, almost all observations of compact smoke bubbles at $20 \, \text{km}$ and above could be attributed to one of the vortices A, B1 and B2. Those which were discarded are under the shape of filaments or tails at altitudes corresponding to the bottom of the bubbles. As in Fig. 1, the location of the ERA5 vortex center is shown as a cross in each panel of Fig. 2. In turn, the location of the smoke bubble centers are shown as square marks in Fig 5 which describes the trajectory of vortex A, and the bars indicate the latitude and altitude extent of the bubble. A perfect match of the horizontal location is not expected as there is no reason for CALIPSO orbit to cross every time the vortex by its center. Nevertheless, we see a very good agreement between the ERA5 trajectory and the location of the 27 smoke bubbles seen by CALIOP that are attributed to vortex A, and the same holds for the two other vortices B1 and B2 (Figs. S2 & S3 of the Supplement) with, both, 21 attributions. The evolution of vortices A, B1 and B2 is also available as animations in Sec. S3 of the Supplement. Vortex A was the first to separate from the mother vortex O by 22

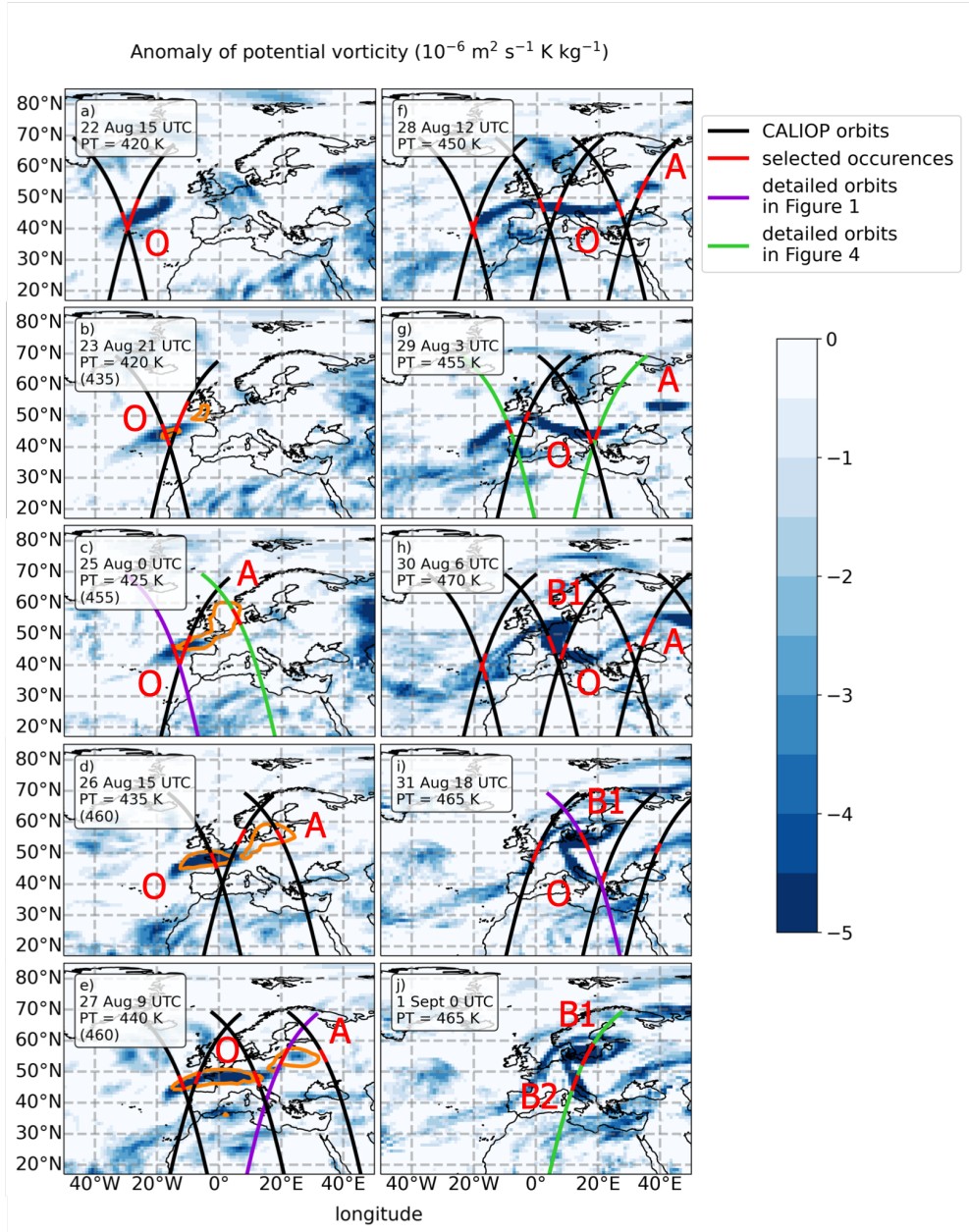

**Figure 3.** Sequence of PV charts showing the splitting of vortex O into vortices A, B1 and B2. The map is plotted on the potential temperature surface corresponding to the core of vortex O or its continuation B2 in the ERA5 tracking. The orange lines are plotted at the isentropic level of vortex A, specified within parenthesis, and show the contour of -2 PVU (1 PVU = $10^{-6}$ $Km^2kg^{-1}s^{-1}$) for vortex A. The black, green and purple lines show the intersecting orbits of CALIPSO and the red segments show the parts of the orbits occupied by the bubble (only the bubble core extent is displayed here). The maps are plotted at the hour matching best the selected CALIOP occurrences. The purple orbits are those corresponding to the sections shown in Fig. 1. The green orbits are those corresponding to the sections shown in Fig. 4. Day-time and night-time orbits are used, the day-time orbit are running from south-east to north-west while the night-time orbits are running from north-east to south-west. Starting from 1st September, the remaining of the vortex O is relabelled as B2.

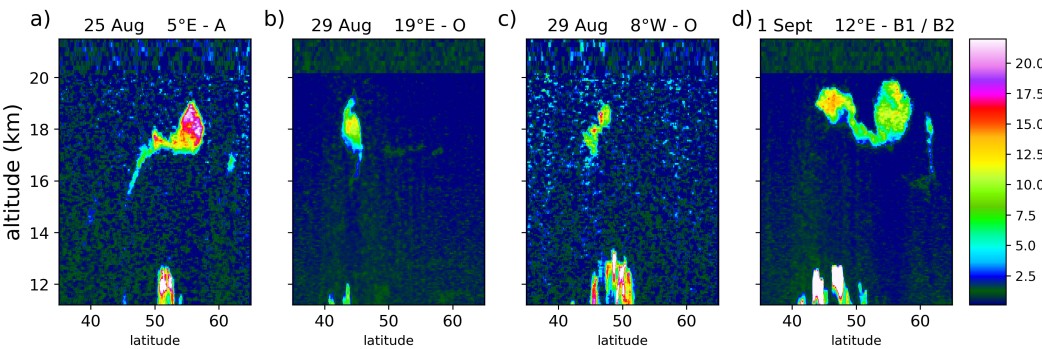

**Figure 4.** Selection of 4 CALIOP scattering ratio sections of the smoke bubbles along the orbits shown in green on Fig. 3. a) day-time section of bubble A on 25 August b-c) east night-time and west day-time sections of the elongated bubble O on 29 August d) night-time double section of bubbles B1 (at ~55° N) and B2 (at ~45° N) on 1st September

August and moved first eastward until it reached central Asia by 1st September at a longitude of 60° E and a latitude of 55° N. It then got trapped into the region of slow motion that extends between the two centers of the Asian monsoon anticyclone (AMA), and started to drift slowly southward while staying at about the same longitude and maintaining its ascent rate. A week later it reached 35° N where is was caught into the easterly circulation and started to move westward crossing Africa and

205 continuing its path which could be tracked until the west Pacific. Figure 6 shows a composite image of successive images of the localized ozone hole from the ERA5. It was easier to track the vortex using the ozone field than the PV. In particular the PV signal almost vanished as it passed over Africa (see the vortex A animation in the Supplement) while the ozone signature was always very clear. We attribute this vanishing to the strong infrared emissivity of the Sahara that limits the sensitivity of the IASI sounders which are important to sense the thermal signature of the vortex (Khaykin et al., 2020). The detection of

210 ozone is less affected as it uses also instruments such as GOME-2 (Khaykin et al., 2020) that operate in the ultraviolet range. The fact that the PV signature re-intensifies as soon as the vortex is over the Atlantic supports this hypothesis. During the last stages of the vortex, by mid-October, we also see a premature loss of the PV signal while the ozone signal is still detectable and can be detected beyond the end of our tracking. This pattern is shared by the two other vortices B1 and B2 and it differs from the 2020 case where such effect is not observed for any of the three vortices studied by Khaykin et al. (2020). Besides the

215 increase of the IASI fleet from two to three instruments, we do not see any drastic change in the observation system between the two events.

While the vortex A was completing its transition to the tropics, the two other vortices B1 and B2 travelled eastward within the westerly flow on the north side of the AMA reaching together the Pacific on 18 September. The vortex B1 crossed the Pacific at mid latitude and got lost near Hudson Bay after crossing most of North America by 14 October. The vortex B2

travelling at higher latitude completed a full round the globe travel during the same period, and got lost over central Asia by 11 October. Figure 7 summarized the trajectories of the vortices O, A, B1 and B2 from their formation to their loss. The total

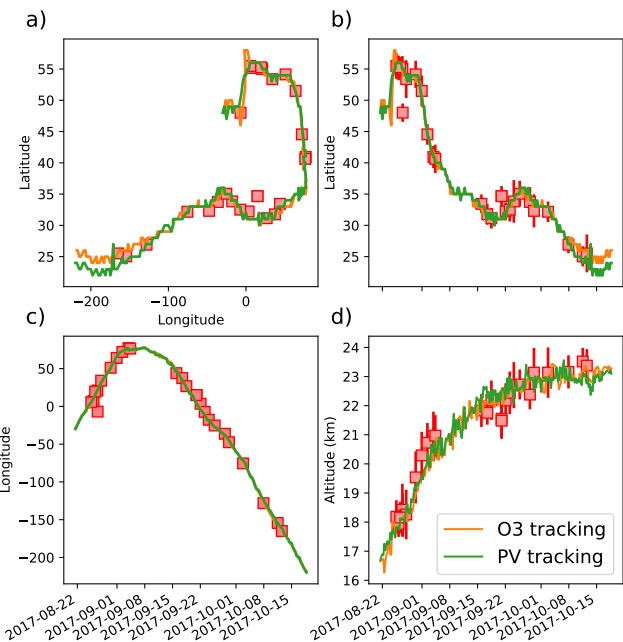

**Figure 5.** Trajectory of the vortex A tracked from the ERA5 fields of PV (orange) and ozone (green). a) Trajectory in the longitude-latitude plane; b, c, d respectively: latitude, longitude and altitude as a function of time. The green curves are mostly masking the orange curves as they almost exactly coincide. The boxes show the location of smoke bubble according to CALIOP during CALIPSO overpasses and the bars indicate the range of the bubble in latitude and altitude.

recorded paths of the four vortices are 13000, 42400, 28500 and 33400 km, respectively. They rose up to 23 km for vortex A and to 21 km for vortices B1 and B2 (altitudes of the core).

## 3.5 Comparison with previous studies

Several previous studies have discussed various stages of the smoke cloud evolution described above although none made the link with a PV structure. A number of comments are here in order:

- The ascent rate was the strongest during the initial stage of vortex O when it rose from its origin just above the tropopause. Previous studies based on CALIOP or limb sounding instruments reported an ascent rate using the upper envelop of the bubble. Yu et al. (2019) reported an ascent from 14 to 20 km from 15 to 24 August and Khaykin et al. (2018) reported an ascent of 30 K of potential temperature per day, that is 3 km per day, between 16 and 18 August. We used the centroid of the PV and O3 anomalies to define the ascent and found (see Fig. S1 of the Supplement) that the vortex O ascended from 12 to 17 km between 14 and 24 August which is consistent with Yu et al. (2019) but we found no trace of a faster ascent. From 24 to 31 August, the two vortices O and A (see Fig. 5) climbed by 2 km and vortex A maintained this rate until 8 September where it reached 21 km. It took one more month to reach the maximum altitude of 23 km while B1 and B2

reached 21 km within the same time. At the very initial stage, Torres et al. (2020) claim an ascent of $20\,\mathrm{K day}^{-1}$ on 14 August when the smoke bubble is detected by the first time in the stratosphere by CALIOP. However, the assumption of a tropopause crossing between 13 August and 14 August is questionable as the sections of CALIOP through the smoke cloud were on its periphery on 13 August, as shown by Fig. 1d of Torres et al. (2020) (even adding the missing night orbit). Therefore, it is equally plausible that the pyro-convective event has injected directly smoke in the stratosphere without need of a large internal radiative heating.

– Peterson et al. (2018) noticed the formation of vortex O from CALIPSO and OMPS / SUOMI NPP overpasses on 14 August in north Canada. Both Khaykin et al. (2018) and Peterson et al. (2018) reported that the first smoke patches reached Europe by 19 August and they were observed by a ground lidar station in central Europe on 22 August (Ansmann et al., 2018; Baars et al., 2019). These patches followed a northern route faster than that of vortex O and were seen as filamentary structures by CALIOP. Their forefront reached Asia by 24 August. By 29 August a thick layer of smoke with aerosol optical depth 0.04 was seen from a lidar in south east France at 19 km (Khaykin et al., 2018) corresponding to the passage of vortex O in the vicinity. According to our tracking, the lidar saw the tail connecting vortex O to vortex B1 in course of separation (see Sect.3.3). Bourassa et al. (2019) noticed the southward motion of vortex A over Central Asia but interpreted it as a split of the plume over this region by 1st September while in fact the separation occurred one week before. It is tempting to see a trace of the vortex A in the OMPS signal at $30°$ N above 20 km during September in Fig.1 of Bourassa et al. (2019) and in the SAGE III low latitude signal above 20 km during October in Fig. 1d of Kloss et al. (2019). Bourassa et al. (2019) have been able to track smoke patches until January 2018 and the global signature at altitudes up to 24 km persisted until summer 2018 (Kloss et al., 2019).

– Kloss et al. (2019), using several satellite instruments and transport calculations, found transport of smoke patches to the tropics that was performed on the eastern southward branch of the AMA but they excluded transport across the AMA which is what we observe for vortex A. These authors focused their work on the events of the last week of August and on levels below 20 km. The transition of vortex A occurred in early September and it was already above 20 km. The smoke layer that Kloss et al. (2019) subsequently followed in the tropics includes the effect of vortex A.

– Baars et al. (2019) show interesting evidence of aerosol patches over Europe and the Mediterranean area reaching 23 km by mid September 2017 and again in mid December 2017. Although unstated, they do not expect the second patches to be the remnant of the first. They instead provide an explanation based on a circuit identified by Kloss et al. (2019), where smoke patches were injected into the tropics by early September. The smoke then rose slowly to higher levels in the tropics and came back to the mid latitudes carried by the Brewer-Dobson meridional circulation. There is, however, a hole in Baars et al. (2019) reasoning which is that the tropical rise is reported to reach 21 km by March 2018 while the aerosols supposedly blown away by the Brewer-Dobson are found at 23 kmover Europe in December 2017. It is now clear that the missing piece of the puzzle is provided by vortex A which reached 23 km by late September (with a top at 24-25 km) leaving a tail along its path.

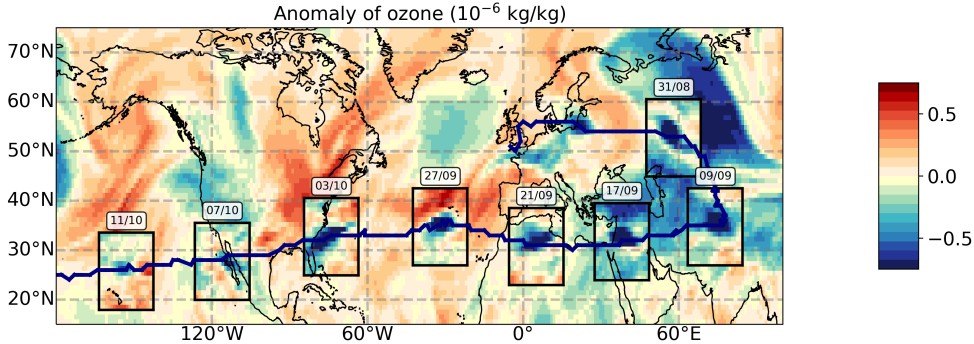

**Figure 6.** Sequence of ozone anomalies along the trajectory of vortex A for selected dates from 31 August to 11 October. The background is the mean ozone anomaly over this period at 460 K. For each selected day, the box shows the ozone anomaly on that day at at 0:00 UTC at the level of the vortex centroid according the the ozone anomaly tracking, reported along the blue line.

## 4  Structure of the vortices in 2020 and 2017

### 4.1  Composite analysis of the vortices in the ERA5

We investigate here the structure of the vortices by the mean of a composite analysis. The dynamical fields surrounding the vortex centroids were regridded regularly in Cartesian geometry in the horizontal and log-pressure altitude (Andrews et al., 1987) in the vertical, within a moving frame which follows the centroids ascent and horizontal displacement. As in Khaykin et al. (2020), the composited model fields were then averaged in time to generate a composite analysis that filters out noise and variability unrelated to the vortices. This procedure also removes short-term vacillations (see Khaykin et al., 2020; Allen

et al., 2020) which are a common property of vortices in shear flow (e.g. Tsang and Dritschel, 2015), thus enabling us to emphasize the mean structure of the vortex. Figure 8 depicts the composite of Lait PV Π (defined in Eq. (2)) and temperature anomaly following vortex A (bottom panels) and the main Koobor vortex (top panels) generated by the 2020 Australian fires and tracked by Khaykin et al. (2020). In the natural coordinates, without expanding the vertical direction, both vortices appear as isolated pancakes of anomalous anticyclonic Lait PV (i.e., negative in the northern hemisphere and positive in the southern

one). The analysed temperature anomaly consists in a vertical dipole surrounding the PV with negative temperature anomaly above and positive temperature below it, where the centers are located on the upper and lower edges of the Lait PV distribution. As noted by Khaykin et al. (2020), this relationship between temperature stratification and vorticity is qualitatively consistent with the thermal wind balance, and characteristic of anticyclonic vortices in the quasi-geostrophic (QG) equations (Dritschel et al., 2004), but also of balanced anticyclones in the full primitive equations (e.g. Hoskins et al., 1985). In the case of the

smoke-charged vortices investigated here, it turns out that their intensity is slightly beyond that of the typical geostrophic flow. The typical Rossby number of the structure can be expressed as follows:

$$Ro = \frac{U}{f\,L_h} \simeq \frac{\bar{\zeta}}{f}, \tag{3}$$

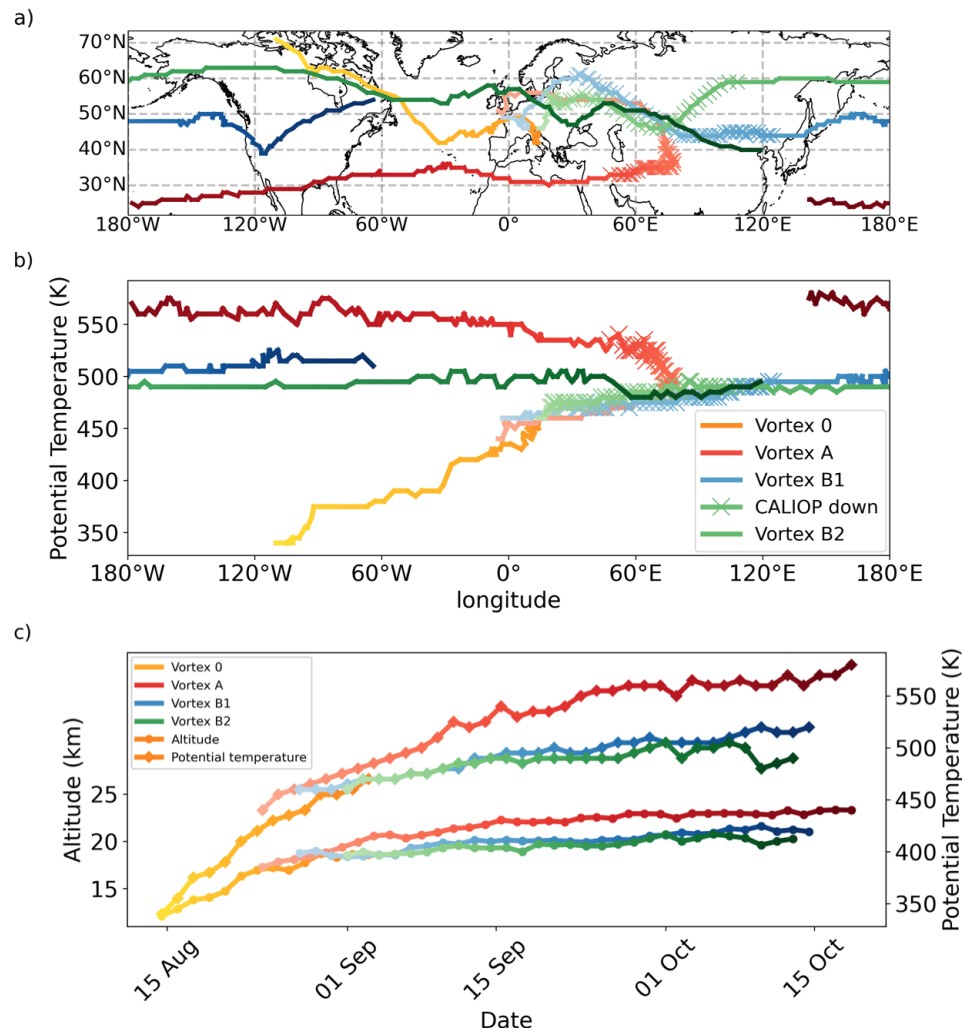

**Figure 7.** Trajectories of the vortices based on low-ozone anomaly from the ERA5 analysis at 6-hour sampling. The colour gradient along each trajectory shows the time evolution of the vortices. We follow vortex O from 14 July 2017 until 31 August where it is relabelled as vortex B2 following its course until 11 October. Vortex A separates from vortex O on 22 August and is followed until 18 October. Vortex B1 separates from vortex O on 27 October and is followed until 14 October. The third panel shows the ascent both in altitude (left axis and lower set of curves) and potential temperature (right axis and upper set of curves). The crosses cover the part of each trajectory where CALIOP was not available.

with $U$ the maximum horizontal wind speed, $L_h$ the horizontal length scale (defined as the diameter of the ring of local wind speed maximum at the altitude of the centroid and estimated in the West-East direction) and $\bar{\zeta} = U/L_h$ the average relative

vorticity. $Ro$ is about $0.06$ in the 2017 case and $0.35$ for 2020. While in 2020 the Rossby number is beyond the typical QG

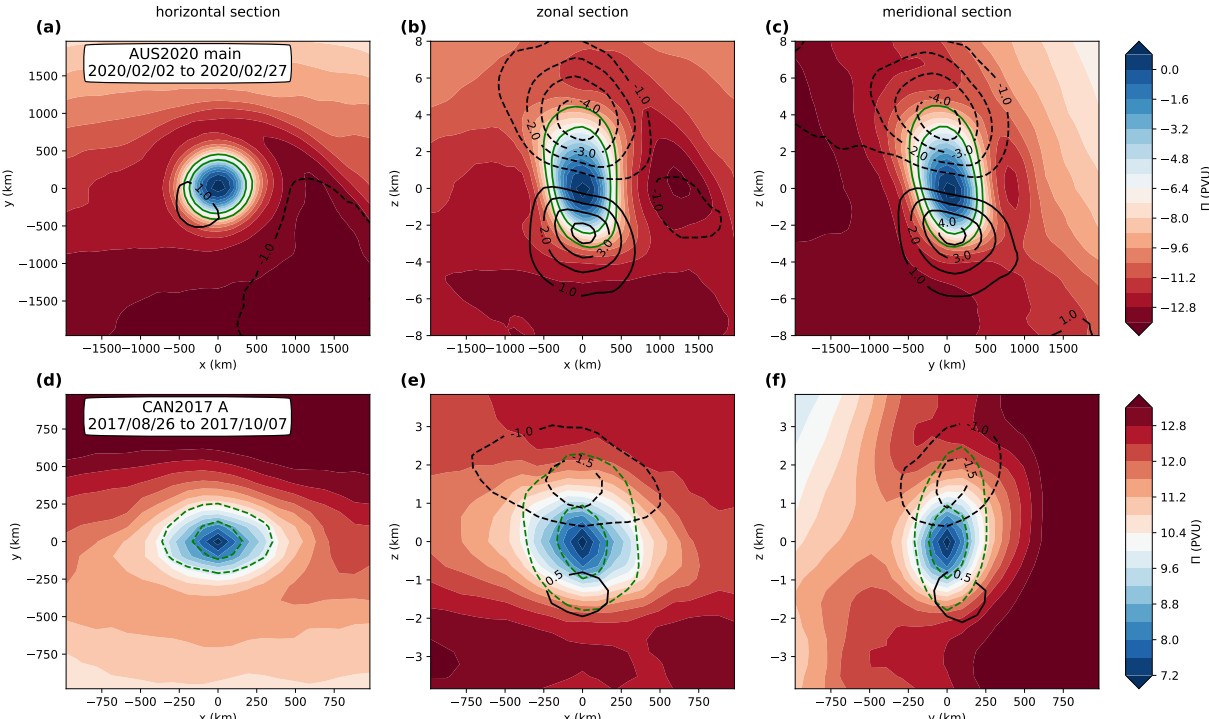

**Figure 8.** Time average composite sections of Lait PV $\Pi$, in PVU (1 PVU = $10^{-6}\,\mathrm{Km^2kg^{-1}s^{-1}}$), following two selected smoke charged vortices, the main Koobor vortex from the 2020 Australian wildfires described in Khaykin et al. (2020) (**a**, **b**, **c**) and vortex A introduced in this paper (**d**, **e**, **f**). Panels **a)** and **d)**, **b)** and **e)** and **c)** and **e)** are respectively horizontal, zonal and meridional sections through the centroid of the vortex. The green lines are contours of anticyclonic vertical vorticity (corresponding to $3 \cdot 10^{-5}$ and $5 \cdot 10^{-5}\,\mathrm{s^{-1}}$ for the top panels, and $-1.5 \cdot 10^{-5}$ and $-2.5 \cdot 10^{-5}\,\mathrm{s^{-1}}$ for the bottom panels). The black contours are temperature anomaly with respect to the zonal mean. Note that the displayed horizontal and vertical ranges are reduced by a factor of 2 for vortex A.

regime ($Ro < 0.1$), the aspect ratio $\alpha$, defined as

$$\alpha = \frac{L_z}{L_h} \simeq 5 \cdot 10^{-3}, \tag{4}$$

($L_z$ being the vertical extent of the contour of vorticity at maximum wind speed at the horizontal location of the vortex centroid) obeys the stratified QG scaling $\alpha \simeq \frac{f}{N}$ in both cases, despite the 2017 vortex A being about 2.3 times smaller in volume than its gigantic 2020 counterpart.

To investigate further the dynamical regime in which the identified vortices evolve (or their representation in the IFS), Table 1 presents their typical sizes, amplitude characterized by the Rossby number $Ro$ and the Froude number:

$$Fr = \frac{U}{L_z N} \tag{5}$$

as well as the absolute vorticity amplitude at the vortex centroid normalized by $f$:

$$\frac{\zeta_a}{f} = \frac{\zeta + f}{f} \qquad (6)$$

which measures the inertial (in)stability of the flow. It should be noted that despite the similarity, $\frac{\zeta_a}{f}$ is not redundant with $Ro$ defined in Eq. (3) and characterizes the extremum rather than the structure average of the vorticity.

Keeping in mind the limited vertical and horizontal resolution of the IFS, it can be noticed that the Froude and Rossby numbers are always of the same order, leading to a Burger number $Bu \sim 1$ as typically encountered in geophysical flows. Furthermore, most vortices in Table 1 obey the QG aspect ratio, so that they are quasi-spherical (see Fig. 8) when the vertical coordinate is stretched by a factor $\frac{N}{f}$. This observation is consistent with numerical studies of ellipsoidal vortices, which have demonstrated that the higher stability of quasi-spherical vortices (Dritschel et al., 2005) and the tendency of aspherical vortices to relax towards sphericity in the stretched coordinate system $\left[x, y, \frac{N}{f}z\right]$ (Tsang and Dritschel, 2015).

Contrary to its aspect ratio, the magnitude of the vortex perturbation is case-dependent and does not necessarily fit in the classical QG scaling, thereby contrasting with the synoptic scale circulation. Indeed, typical vortex-averaged Rossby numbers range from $0.06$ up to $0.35$, a range similar to observed mesoscale and submesoscale oceanic eddies (e.g. Le Vu et al., 2017). For the intense 2020 Koobor vortex, the maximum vorticity in the vortex core is on the verge of inertial instability or even slightly beyond its threshold for linear flows (Hoskins, 1974), as can be seen from the positive $\Pi$ values in Fig. 8 (see also the small negative values of $\frac{\zeta_a}{f}$ in Table 1). This property was maintained over one month and half from the formation of the Koobor vortex to its first breaking (see Fig. S4 of the supplement). Although it is not clear how realistic the ECMWF vortices are, their amplitude is likely underestimated since they are forced by data assimilation and the standalone model does not simulate them. We can therefore not generally conclude on the amplitude of the perturbations, in particular the smaller Northern Hemisphere vortices. Nevertheless, the 2020 cases demonstrate that the vorticity anomaly may reach the threshold for inertial instability at which it likely saturates.

Related to their different magnitudes, the vortices also have distinct impacts on their immediate surroundings, as can be identified in Fig. 8. While the 2020 Koobor vortex was strong enough to generate a significant cyclonic PV anomaly eastward of the vortex centroid which rolls up around it on its northern edge, no such signature can be distinguished around vortex A. In a zonal plane, the 2020 negative PV patch has diameter comparable to the Koobor vortex but reduced magnitude. The existence of this negative anomaly can be attributed to the equatorward advection of cyclonic PV on the eastern flank of Koobor and is likely responsible for the equatorward beta-drift (Sai-Lap Lam and Dritschel, 2001) undergone by Koobor during its ascent.

Finally, for completeness, two further remarks should be made. First, note that the vertical temperature dipole is slightly asymmetric with larger magnitude of the negative temperature anomaly. Second, panel **(c)** of Fig. 8 shows that the vertical axis of Koobor exhibits on average a small tilt with altitude, being slanted along the South-North direction, which is perpendicular to the prevailing background shear. This property is a common characteristic of vortices undergoing shear (Tsang and Dritschel, 2015) and related to the temporal vacillations of Koobor described by Allen et al. (2020) and also seen in Fig. 6 of Khaykin et al. (2020).

**Table 1.** Characteristics (horizontal diameter $L_h$, vertical depth $L_z$, aspect ratio $\alpha$, QG aspect ratio imposed by the environment $f/N$, Rossby number, Froude number and maximum vorticity $\frac{\zeta_a}{f}$) of the six smoke charged pancake vortices originating from the Canadian (2017) and Australian (2020) wildfires. The 2020 Koobor vortex was long-lasting and is here decomposed into two periods.

| name | geographic origin | considered period | $L_h$ (km) | $L_z$ (km) | $10^3\alpha$ | $10^3\frac{f}{N}$ | $Ro$ | $Fr$ | $\frac{\zeta_a}{f}$ |
|------|-------------------|-------------------|------------|------------|--------------|-------------------|------|------|---------------------|
| Koobor | Australia | 2020/01/07 to 2020/01/27 | 784 | 6.1 | 7.8 | 6.6 | 0.35 | 0.30 | 0.03 |
| Koobor | Australia | 2020/02/02 to 2020/02/27 | 784 | 6.1 | 7.8 | 5.8 | 0.33 | 0.25 | -0.09 |
| 2nd Vortex | Australia | 2020/01/18 to 2020/01/27 | 588 | 3.8 | 6.5 | 6.6 | 0.35 | 0.35 | -0.12 |
| 3rd Vortex | Australia | 2020/01/20 to 2020/02/07 | 588 | 7.7 | 13.1 | 7.9 | 0.10 | 0.06 | 0.72 |
| Vortex A | Canada | 2017/08/26 to 2017/10/07 | 686 | 3.5 | 5.1 | 4.8 | 0.06 | 0.06 | 0.6 |
| Vortex B1 | Canada | 2017/08/27 to 2017/10/07 | 588 | 4.5 | 7.6 | 6.2 | 0.12 | 0.10 | 0.56 |
| Vortex B2 | Canada | 2017/09/01 to 2017/10/07 | 490 | 3.8 | 7.8 | 7.0 | 0.11 | 0.09 | 0.6 |

## 4.2 Diagnosing diabatic tendencies

By comparing the 2020 Koobor vortex in ECMWF analyses and model forecasts, Khaykin et al. (2020) have shown two systematic shortcomings of the free-running model with respect to the analysis, namely:

- a failure to reproduce the observed ascent of the structure at about $6\,\mathrm{K day^{-1}}$ in potential temperature: on the contrary, the modeled anticyclones tend to remain at constant $\theta$;

- a decay of the vorticity anomaly within one week, while the observed vortex survives for more than 3 months.

This suggests that data assimilation of observed temperature and ozone profiles is a necessary ingredient to both ascent and the maintenance of the vortex in the model, while some physical or spurious processes act to dissipate the structure. In the atmosphere, it is assumed that the forcing is exerted by radiative heating through solar absorption by black carbon aerosols within the smoke (Yu et al., 2019). Increments are thus expected to play substitute in the analysis system for the diabatic tendency maintaining the vortices, as the wildfires smoke is missing in the model.

### 4.2.1 Temperature and vorticity tendencies

Figure 9 (**a**, **d**) depicts the average composite of the ERA 5 total heating due to physics calculated over the forecast. This field is dominated by the longwave radiative heating component since the shortwave absorption by the smoke is missing from the IFS. The dominant feature shown by Fig. 9 is a damping of the temperature anomaly $T'$ with respect to radiative equilibirum temperature which can be cast into a Newtonian relaxation as:

$$\frac{\mathrm{D}T'}{\mathrm{D}t} \simeq -\frac{T'}{\tau_{rad}} \tag{7}$$

where $\tau_{rad}$ is the radiative damping rate. Figure 9 suggests $\tau_{rad} \simeq 6-7\,\text{days}$. This is consistent with the life-time of the structure in the ECMWF forecast which is about one week (Khaykin et al., 2020), suggesting that radiative dissipation plays a major role in the decay of the vortex in the model but also in the real atmosphere.

However, since the model itself cannot sustain the vortices, it comes to data assimilation to maintain the structure. It was demonstrated by Khaykin et al. (2020) that the IFS extracts information from the thermal signature and the ozone anomaly detected by satellite instruments. The temperature increments are shown in Fig. 9 (**b, e**). In contrast with the heating rates, which are instantaneous forcing of the temperature by physical processes, the temperature increments result from the balanced dynamical response of the temperature field to the forcing induced by the discrepancy of measured radiances with respect to their estimated values in the assimilation procedure. Hence, the temperature pattern does not match the observed aerosol bubble collocated with the vortex which would be expected for the shortwave aerosol absorption. It rather exhibits a multipolar structure, essentially oriented vertically. In the 2017 case with limited vertical ascent rate, the $T$ increment vertical dipole mainly cancels the radiative damping. In 2020, it is rather a tripole which enables the maintenance but also the ascent of the dipolar temperature anomaly structure.

Due to this dynamical adjustment, temperature is not the only field incremented via data assimilation. Figure 9 (**c,f**) shows the increments of relative vorticity $\zeta$. They contribute again to the maintenance (2017 case) and ascent and maintenance (2020 case) of the vortices. Moreover, similarly to the composite vortex, the composite vorticity and temperature increment tendencies exhibit the structure of a balanced vortex: the 2017 anticyclonic $\zeta$ tendency monopole is sandwiched between the two extrema of temperature increment dipole whereas in 2020 the tripole of $T$ increment alternates with the two layers of vorticity anomalies. Overall, the $T$ and $\zeta$ patterns of the 2020 assimilation increment are qualitatively consistent with the expected effect of a localized heating in a rotating atmosphere (as sketched in Fig.10 of Hoskins et al., 2003).

Together with the apparently balanced vortex structure of the increments, the similarity with the theoretical response in Hoskins et al. (2003) suggests that potential vorticity (or Lait PV) is the appropriate field to consider for a more straightforward interpretation of the increments in terms of missing diabatic processes. This is the focus of the next subsection.

### 4.2.2 Lait PV and ozone tendencies due to assimilation increments

Composite time average increments of Lait PV $\Pi$ are presented in Fig. 10. As stressed above, $\Pi$ increments result from a combination of both temperature and vorticity and appear due to the implicit incorporation of temperature observations into an approximately dynamically-balanced response by the 4D-Var scheme used in the ERA5.

Contrary to the mean vortex structure in Fig. 8, we notice a stark contrast between the Northern and Southern Hemisphere vortices. In the 2017 case, the $\Pi$ increment shows a dominant monopole structure whose extremum is slightly shifted upward (by about $500\,\text{m}$) with respect to the $\Pi$ extremum (Fig. 8). The effect of the increment is thus mainly to reinforce the existing $\Pi$ anomaly and, secondarily, to drive its ascent. In contrast, the 2020 case shows essentially a vertical dipole structure which forces an ascent of the $\Pi$ center. The dipole is furthermore tilted along a north-west to south-east axis. A qualitative analysis (Appendix B) shows that the observed westward tilt of the increment dipole emerges as a consequence of the negative zonal wind shear encountered by the 2020 Koobor vortex along its ascent. More precisely, it is the combination of this shear and the

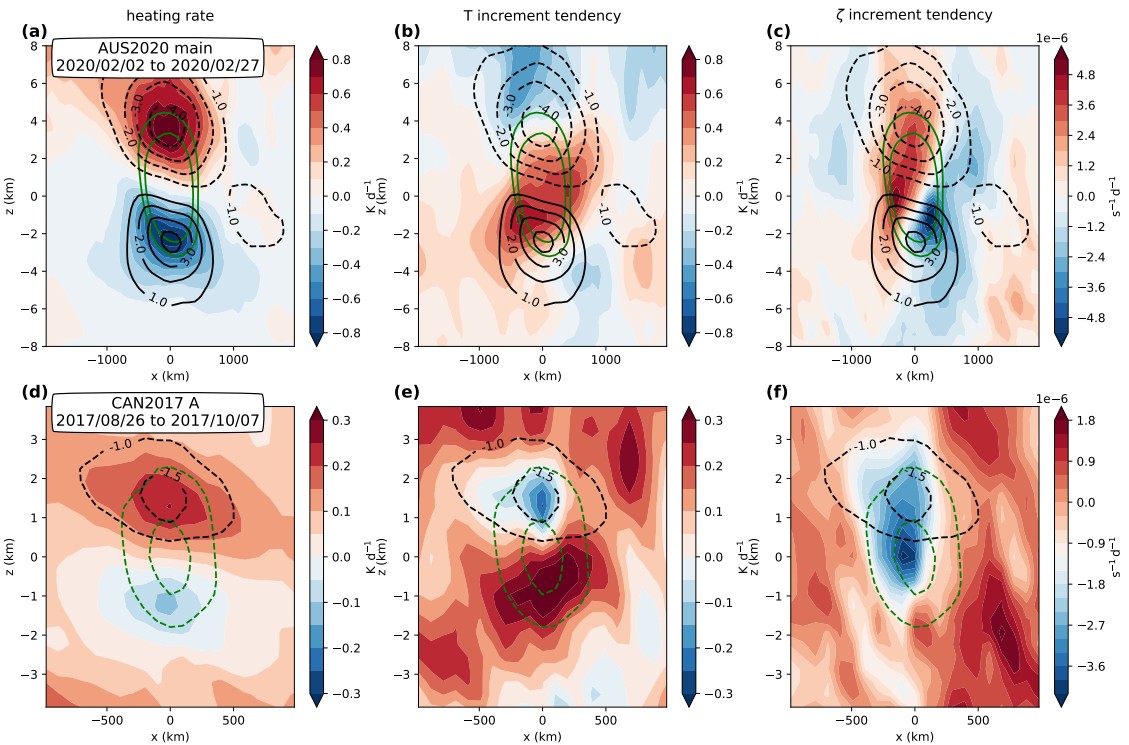

**Figure 9.** Time average composite sections of ERA 5 total heating rate (**a**, **c**), increment-induced temperature (**b**, **e**) and vorticity tendency (**c**, **f**) following two selected smoke charged vortices, the major vortex from the 2020 Australian wildfires described in Khaykin et al. (2020) (**a**, **b**, **c**) and vortex A (**d**, **e**, **f**). The green lines are contours of anticyclonic vertical vorticity (corresponding to $3 \cdot 10^{-5}$ and $5 \cdot 10^{-5}$ s$^{-1}$ for the top panels, and $-1.5 \cdot 10^{-5}$ and $-2.5 \cdot 10^{-5}$ s$^{-1}$ for the bottom panels). The black contours are temperature anomaly with respect to the zonal mean. Note that the displayed horizontal range is reduced by a factor of 2 for vortex A.

distribution of the increment tendency over the 12 hours analysis time window which is responsible for the tilt. This feature may also be seen in other periods of Koobor lifetime during which the structure was drifting eastward, such as in the middle
of January 2020 (see Fig. B1), emphasizing that the inclination of the increment dipole is due to the background wind shear rather than the local wind at the altitude of the vortex.

The different patterns of Lait PV $\Pi$ (Fig. 8) and its increment (Fig. 10) around the vortex are mimicked in the ozone field, as shown in Fig. 11. As described above, the ascending anticyclonic vorticity anomalies are accompanied by negative ozone anomalies (Fig. 11 a and c). Compared with Figs. 8 and 10, Fig. 11 shows that the patterns of the ozone anomaly bubble and its
increment are very similar to those of $\Pi$. We note that the magnitude of the increments may vary depending on the period chosen for the composite, for instance due to the different ascent speed of the vortex in log-pressure altitude coordinate ($250\,\mathrm{m\,day}^{-1}$

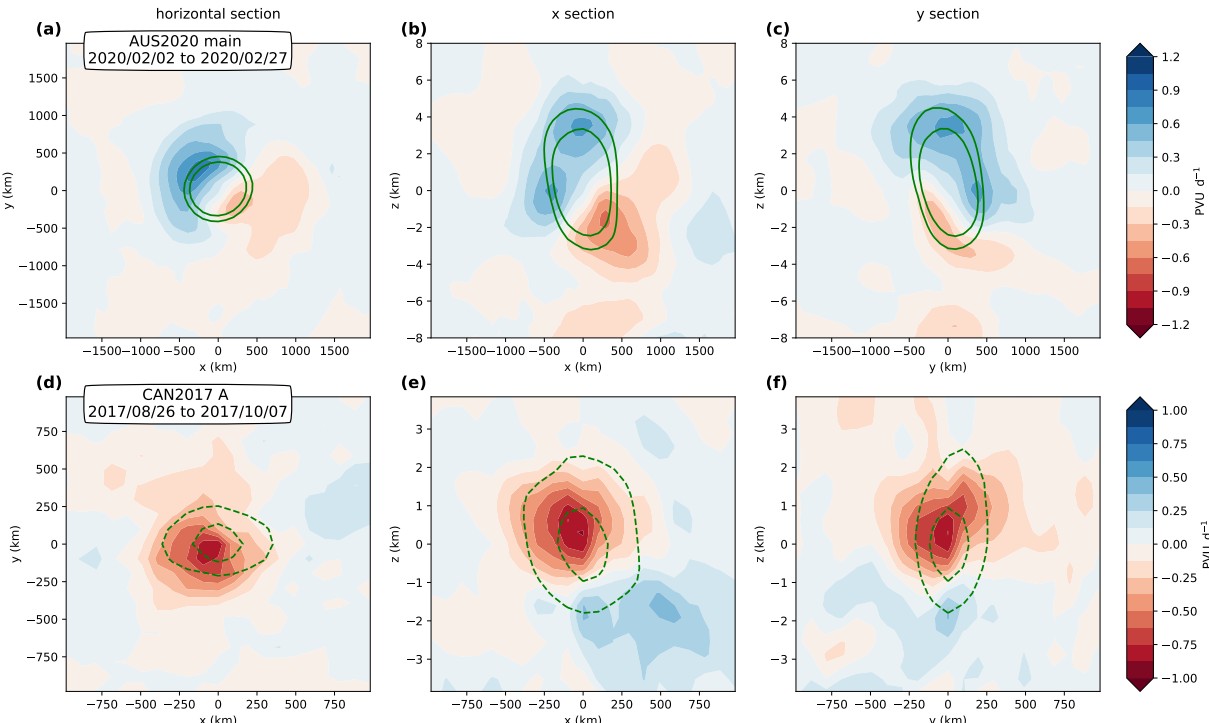

**Figure 10.** Same as Figure 8, but for Lait PV increments.

in January versus $150 \, \mathrm{mday}^{-1}$ in February for Koobor). However, the monopole and tilted dipole structures shown in Fig. 10 and Fig. 11 are representative of the typical situation found. Overall, model increments tend to (i) counter the dissipation and (ii) drive a cross-isentropic vertical motion of PV and ozone anomalies.

At this point, it is tempting to interpret the low ozone, low absolute PV anticyclones as resulting from the vertical advection of smoke-charged tropospheric air bubbles conserving both their low absolute PV and their tropospheric tracer content during their ascent in the stratosphere. In the case of the 2020 Koobor vortex, this perception is broadly consistent with the behavior of PV and PV anomaly at the center of the vortex, which respectively remain constant or exhibit a steady increase related to the large vertical gradient of PV within the stratosphere (Fig. S4 of the supplement). It is also consistent with our analysis of the
developing stage of the mother vortex O in Sec. 3.2. We note, however, that the analogy between inert tracer and PV transport arises from their Lagrangian conservation in the adiabatic inviscid case and breaks in the presence of diabatic processes (Haynes and McIntyre, 1987). The maintenance of this relationship hence calls for dedicated theoretical and numerical investigations.

## 5    Conclusions

The generation of smoke-charged vortices rising in the southern stratosphere was discovered following the Australian wildfires
at the turn of 2019 (Khaykin et al., 2020). We find here that similar events took place in 2017 in the northern stratosphere after

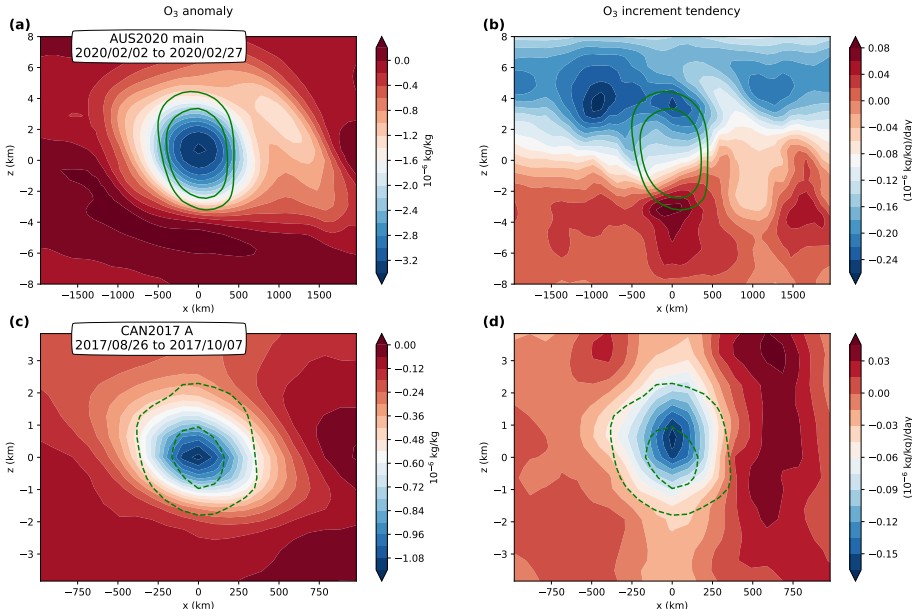

**Figure 11.** Same as Figure 9, but for ozone anomalies from the zonal mean ((**a**), (**c**)) and ozone increments ((**b**), (**d**)).

the British Columbia wildfires that culminated in early August 2017 when a large plume of smoke and of low potential vorticity air has been injected in the lower stratosphere on 12 August 2017. We show that soon after, a vortical structure developed in the plume and started to rise in the stratosphere. During the following days this smoke vortex became an isolated bubble and was transported eastward across the Atlantic while getting elongated by the wind shear and reaching $19\,\mathrm{km}$ in altitude at its
top. Subsequently, that structure was split over western Europe into three separate vortices that could be followed until the middle of October 2017. Two of them kept moving eastward in the middle latitudes and performed a round the globe travel, rising up to a potential temperature of $530\,\mathrm{K}$ ($21\,\mathrm{km}$). The third one has transited to the subtropics across the Asian monsoon anticyclone and moved eastward until Asia. It also rose higher, to $570\,\mathrm{K}$ ($23\,\mathrm{km}$).

Our study demonstrates again the ability of the advanced assimilation system used at the ECMWF which is the basis of
the ERA5 to exploit the signature left by smoke vortices in the temperature and ozone fields and reconstruct the balanced vortical structure. The balance is produced during the iterations of the assimilation and therefore the assimilation increment reflects this balance by providing an increment on the wind as well as on the temperature. The increment is consistent with the expected effect of a localized heating due to the radiative absorption of the smoke. It fights the longwave radiative dissipation and moves the vortices upward. It is quite likely, however, that the detection of vortices by the assimilation system is limited
by the sensitivity of the satellite instruments to the disturbances in the temperature and ozone fields. There are also detection limits in CALIOP and limited chances to overpass small-scale structures if they are sparse enough. It is therefore possible that a number of small-scale vortices escape the direct methods used in this study. It is also possible that dynamical constrains, like the ambient shear, limit the existence of such vortices. It is thus difficult to conclude from our study what is the generality of

such structures and their global impact. It is however quite clear that they provide an effective way for smoke plumes to stay compact and concentrated inducing a rise of the order of $10\,\mathrm{km}$ or more in the stratosphere that, in turns, increases the life-time and the radiative impact mentioned in previous studies (Ditas et al., 2018) from months to years knowing the character of the Brewer Dobson circulation (e.g. Butchart, 2014).

Long lived anticyclones dubbed as "frozen-in-anticyclones" (FrIAC) by Manney et al. (2006) have already been reported in the Arctic stratosphere. They share a long life span with the smoke anticyclones but they differ in many other respects. The FrIAC are deep barotropic structures extending from $550\,\mathrm{K}$ to $1300\,\mathrm{K}$ and are observed in the polar region after the breaking of the winter polar vortex. They are generated by the isentropic intrusion of low latitude air with low PV (Allen et al., 2011; Thiéblemont et al., 2011). Instead, the smoke vortices exhibit a strong baroclinic structure with a vertical temperature dipole and are propelled through the layers of the stratosphere by their internal heating bringing tropospheric air to the middle stratosphere at latitudes where the Brewer circulation does the opposite (Butchart, 2014).

As volcanic plumes in the stratospheric are usually made of secondary sulfuric acid aerosols that are considerably less absorbing than the black carbon, raising compact plumes are not expected to be seen in the stratosphere after extratropical volcanic eruptions. Such a case, however, has been reported after the 2019 eruption of the Raikoke (Chouza et al., 2020; Muser et al., 2020). This event which displays a slow rise and is not associated with a vortex in the ERA5 might fall below the detection limit but opens the question of the possible role of heating by volcanic aerosols.

The conditions to maintain the stability of the smoke vortices and those leading to their final dilution are also not yet understood and this work is only opening a path to be explored.

## Appendix A: Tracking of the smoke bubbles

Figure A1 shows two screenshots as examples of the interactive tracking of the smoke bubbles in CALIOP sections of the scattering ratio. The first case (Fig. A1a) is taken from bubble O on 19 August when it was elongated within the Atlantic trough. The second case (Fig. A1b) is taken from bubble A on 24 September while it was moving westward over the Atlantic. The interactive method used to perform the tracking is displayed in an animation in Sec. S1 of the Supplement.

## Appendix B: Increment structure around the ascending vortex in a sheared background wind

The average composite of Lait PV increment for the 2020 anticyclone (Fig. 10) exhibits a tilted dipole structure, whose inclination by far exceeds the slight dip observed for the vortex axis and figured by the green vorticity contours in Fig. 10 **b**. This tilt causes the dipole to appear not only in vertical panes but also in the horizontal one (Fig. 10 **a**). The existence of the tilt is not related to the direction of the drift of the vortex: in January, when Koobor is drifting eastward due to the prevailing Westerlies, we observe the same general orientation of the increment dipole, as shown in Fig. B1.

From a purely kinematic point of view, the dipolar structure of the increment shows that the forecast model systematically underestimates both upward and westward motions (as well as northward in Fig. B1). While it is unlikely that the ECMWF

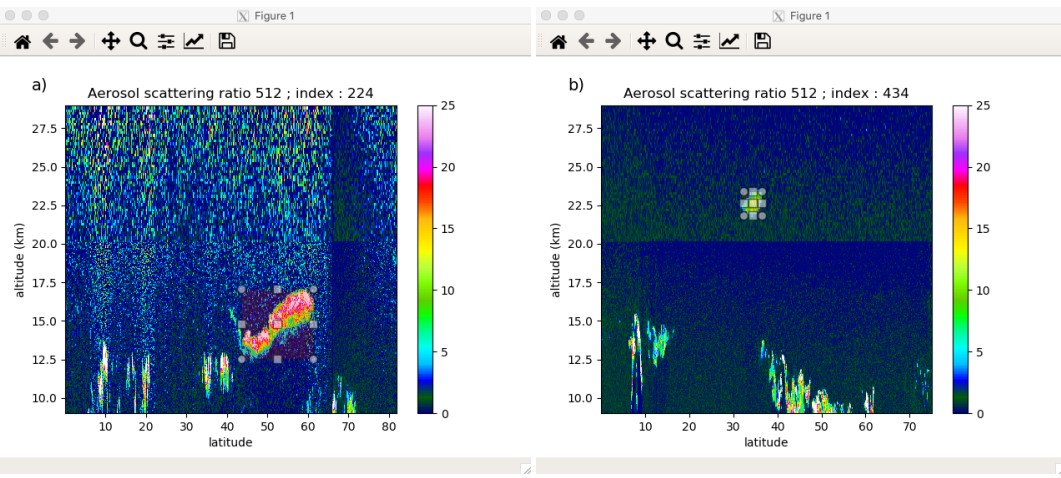

**Figure A1.** CALIOP sections with the superimposed box showing how the tracking is performed a) day-time orbit over bubble O on 19 August, b) night-time orbit over bubble A on 24 September.

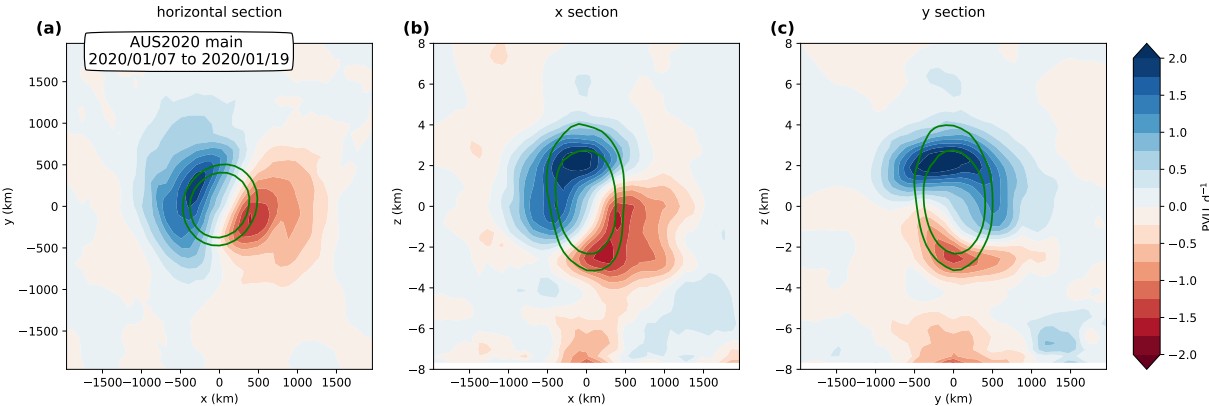

**Figure B1.** Same as Figure 10, but for Koobor Lait PV increments during the 2020/01/07-2020/01/19 time period. Note the different colorbar compared to Fig. 10. The different magnitude of the increments is due to the different ascent speed of the vortex in log-pressure altitude coordinate (250 m/day in January versus 150 m/day in February).

forecast model underestimates zonal wind magnitude over such a long period, this missing westward displacement filled by the increments may be attributed to missing vertical ascent in an easterly sheared background wind.

This can be grasped by comparing trajectories advected by a sheared flow (wind profile $U(z) = U_0 + \Lambda z$ where $\Lambda$ is the shear) with or without an ascent. The two situations are contrasted for a simplified case in Tab. B1. Over the finite analysis time window $\Delta t$, the vertical ascent rate $W$ causes the parcel (here, the vortex) to be advected in a region of different mean

wind and drift away from its non-ascending cousin. Hence, the additional motion brought by the ascent has both vertical and

**Table B1.** Analytical trajectory of a bubble in an idealized sheared wind profile: $U(z) = U_0 + \Lambda z$ with (forecast) or without (analysis) vertical ascent, and resulting average speed.

| | forecast | analysis (atmosphere) |
|---|---|---|
| vertical velocity | $\dot{Z} = 0$ | $\dot{Z} = W$ |
| horizontal velocity | $\dot{X} = U_0$ | $\dot{X} = U_0 + \Lambda Z$ |
| vertical trajectory | $Z(t) = Z_0$ | $Z(t) = Z_0 + Wt$ |
| horizontal trajectory | $X(t) = X_0 + U_0 t$ | $X(t) = X_0 + U_0 t + \frac{\Lambda W}{2} t^2$ |
| averaged vertical velocity over $\Delta t$ | $\frac{\Delta Z}{\Delta t} = 0$ | $\frac{\Delta Z}{\Delta t} = W$ |
| averaged horizontal velocity over $\Delta t$ | $\frac{\Delta X}{\Delta t} = U_0$ | $\frac{\Delta X}{\Delta t} = U_0 + \frac{\Lambda W}{2} \Delta t$ |

horizontal components, which can be expressed as:

$$(\dot{X}, \dot{Z})_{incr} = \left( \frac{\Lambda W}{2} \Delta t, W \right) \tag{B1}$$

The tendency of $\Pi$ associated with a translation at speed $(\dot{X}, \dot{Z})_{incr}$ is:

$$\partial_t \Pi = -\dot{X}_{incr} \partial_x \Pi - \dot{Z}_{incr} \partial_z \Pi \tag{B2}$$

If we consider the isolated bubble of maximum potential vorticity anomaly $P_{max}$ and length scales $L_x$ and $L_z = \alpha L_x$, the magnitude of the tendency required to translate it along $x$ and $z$ are then:

$$\partial_t \Pi|_{x=0} = -W \frac{2 \Pi_{max}}{L_z}$$
$$\partial_t \Pi|_{z=0} = -\Lambda W \frac{\Delta t}{2} \frac{2 \Pi_{max}}{L_x} \tag{B3}$$

and their ratio $\gamma$ is:

$$\gamma = \frac{\partial_t \Pi(z=0)}{\partial_t \Pi(x=0)} = \alpha \Lambda \frac{\Delta t}{2} \tag{B4}$$

In the case of the 2020 vortex, $\alpha \simeq 7.8\,10^{-3}$ and the background shear $\Lambda$ estimated in a $8\,\mathrm{km}$ layer around the vortex varies from $\simeq -3.\,10^{-3}\,\mathrm{s}^{-1}$ in January 7-19 to $\simeq -2.\,10^{-3}\,\mathrm{s}^{-1}$ in February 2-27. With those values and the assimilation window $\Delta t = 0.5\,\mathrm{day}$, that $|\gamma|$ is of the order of 0.3 to 0.5, which is quantitatively consistent with Fig. 10, panel **b)** and Fig. B1. When the shear is properly taken into account, the westward drift induced by the increments can thus be reconciled with the idea that the ascent is the main process lacking from the model. Note that this reasoning also applies to ozone.

*Author contributions.* The four authors performed together the analysis. HL, BL and AP wrote the manuscript. All authors contributed to the final version.

*Competing interests.* We declare that no competing interest are present.

*Acknowledgements.* CALIOP data were provided by the ICARE/AERIS data centre. ERA-5 data where provided by Copernicus Climate Change Service Information. This work was supported the TTL-Xing ANR-17-CE01-0015 project of Agence Nationale de la Recherche.

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
