# Peer review of "Smoke-charged vortices in the stratosphere generated by wildfires and their behaviour in both hemispheres: comparing Australia 2020 to Canada 2017"

_Atmospheric Chemistry and Physics, 2020_

## Referee Comment (RC1) · Anonymous Referee #1 · 10 Dec 2020

**Review of Lestrelin, H., B. Legras, A. Podglajen, and M. Salihoglu, Smoke-charged vortices in the stratosphere generated by wildfires and their behaviour in both hemispheres: comparing Australia 2020 to Canada 2017, Atmos. Chem. Phys. Disc., 2020**

**General Comments**

This paper details the dynamics of a newly-discovered phenomenon associated with large pyroCb smoke plumes in the stratosphere, namely the self-organized anticyclonic vortices that are formed due to absorption of solar radiation by black carbon within the plumes. A comparison is made between the 2019-2020 Australian plumes and several plumes associated with large Canadian fires in 2017. Detailed analysis of the Canadian plumes use the Lait Potential Vorticity (PV) from ERA5 to track the vortex evolution and to examine the composite dynamical structures of the vortex. ERA5 ozone is also used to successfully track the vortex locations. Composites of PV, temperature, and ozone help to further understand these features. The paper also details how these features are maintained in the analyses by assimilation of temperature and ozone data, and it examines various dynamical indices to test whether the vortices are in balance and/or inertially unstable. This paper provides an excellent addition to the study of smoke-induced dynamics.

**Specific Comments**

Line 10: You use the term "low absolute potential vorticity" here. Just to be clear, does this mean low magnitude (i.e., absolute value) of the potential vorticity?

Line 83: I don't think "g is the free-fall acceleration" is necessary, since "g" isn't in Eq. (1).

Lines 112-114: Quoting from the paper, "we used both $\Pi$ and the ozone anomaly defined as the deviation with respect to the mean at the same latitude and altitude." So are both $\Pi$ and ozone defined using the anomaly with respect to the zonal mean, or is just ozone calculated as the anomaly, while $\Pi$ is the raw value from Eq. (2)? This sentence could be read either way.

Lines 112-114: As mentioned by the authors, the commonly used PV has a disadvantage of large background vertical gradient. While Khaykin et al. (2020) used relative vorticity, Kablick et al. (2020) used the "regular" PV to analyze the 2019-2020 Australian smoke plumes. They used the PV anomaly relative to the zonal mean in units of percent of the absolute value of the zonal mean PV. This alternate approach also reduces the influence of the large background gradient.

Line 139-142: You talk about the "kernel of almost zero PV and low ozone...." To visualize this statement, it would be interesting to see latitude/altitude cross-sections of PV and ozone along the CALIPSO track to compare with Figures 1 and 2.

Line 147 (also Lines 158 and 394) : I am unfamiliar with the term "thalweg". Are you referring to a trough? Could you provide some dynamical field on the maps to indicate where this is occurring to help visualize the point you're making?

Line 194-195: Is the statement "it differs from the 2020 case where such effect is not observed for any of the three vortices" referring to results in Khaykin et al. (2020)?

Section 3.5. This section provides a helpful reference of other papers that have studied this event. The PV anomaly associated with smoke during the Canadian event was also examined in a recent paper by Allen et al. (2020). While that paper focused primarily on the "Koobor" vortex, they also included a PV anomaly map for 28 August 2017 associated with what this paper calls "Vortex A" (see Figure 16 of the following reference).

> Allen, D. R., M. D. Fromm, G. P. Kablick III, and G. E. Nedoluha, 2020: Smoke With Induced Rotation and Lofting (SWIRL) in the Stratosphere, J. Atmos. Sci., 77, 4297-4316, https://doi.org/10.1175/JAS-D-20-0131.1.

Line 243-245: It is interesting that the 2017 case doesn't show the temperature dipole. Is that simply due to the contours chosen (i.e., the warm anomaly is really there, but it is less than 1.0 K)?

Line 252: It is unclear exactly how you calculate the horizontal length scale $L_h$. The text says it is defined as "the diameter of the ring of wind speed maximum". Do you calculate the diameter from the wind speed explicitly for each case? If so, does the wind speed calculation involve removing the background wind in order to focus on the wind associated with the anticyclone?

Line 256: Similarly, $L_z$ is "the vertical extent of the vorticity contour at maximum wind speed". Is this calculated explicitly? Would it be possible to include some more details on this calculation?

Line 257: You say "the 2017 vortex A being about 8 times smaller in volume than its gigantic 2020 counterpart". This difference seems too large. If the cloud is considered as a cylinder, then volume is $V=\pi(L_h/2)^2 L_z$. Using numbers from Table 1 we get $V_A = \pi(686/2)^2 3.5 = 1.3\times10^6$ km$^3$, and $V_{Koobor} = \pi(784/2)^2 6.1 = 2.9 \times10^6$ km$^3$. So Koobor is only 2.9/1.3=2.2 times larger in volume than vortex A. Are these estimates correct, or do you use another method to estimate volume?

Line 263: You may want to define the condition for inertial instability here.

Line 328-330: It looks like the dates used for the Australian vortex are 2-27 February in Figure 8 (mean structure), Figure 9 (heating rate, temperature and vorticity tendencies), and Figure 11 (ozone tendencies), but for Figure 10 (PV increments) the dates are 7-19 January. It there a particular reason that different dates are chosen for PV? Are the mean composites and increments

of PV much different if you calculate them for the different periods? Also, are the green lines on Figure 10 from 7-19 January or 2-27 February?

Line 335: The northwest-southeast tilt of the PV increments for the Koobor vortex, shown in Figure 10a, is interesting. In the recent analysis by Allen et al. (2020), they examined how Koobor tilts with height and found a NW-SE tilt of the vortex in January. They used a dynamical argument to show how this tilt may develop from internal vortex dynamics in a shear flow. The PV increments shown in this paper appear to support this observed dynamical structure. Also, would the same argument you make in Appendix B apply to the ozone structure seen in Figure 11?

Line 344: Does "low absolute PV" mean low magnitude (i.e., low absolute value of PV)?

Line 408: May want to define terms explicitly in the text here, particularly W and $\Lambda$. I assume these are vertical wind and vertical shear of the zonal wind. They are indirectly defined in the Table B1, but not in the text.

Line 416: How is the wind shear estimate calculated here (i.e., what time range is used)?

Figure 3 caption: Could you include in the caption what time of day was used for the PV analyses?

Figure S4: Is this calculated with normal PV or with the Lait PV? Also, as a reference, it would be useful to include the zonal mean PV on this plot. This should become increasingly negative with time as the parcel ascends. Do the Canadian plumes show a similar behavior?

I enjoyed the animations in the supplementary material. I assume the PV used in the animations is the normal PV, not the Lait PV, right?

Lastly, there are quite a few different names used in this paper for different aspects of this new phenomenon. For example the terms "smoke-charged vortex", "smoke charged pancake vortex", "smoke vortex", "smoke plume", "smoke bubble", and just "bubble" could possibly be condensed into fewer descriptions. The Australian plumes are called "Koobor", "2nd Vortex", and "3rd Vortex", while the Canadian plumes are "Vortex O" (also called "mother vortex"), "Vortex A", "Vortex B1", and "Vortex B2". Different terms are also used for "Koobor", such as "main vortex", and "major vortex". Given this is such a new discovery, to avoid potential confusion, terms could be consolidated and defined (e.g., how does the term "bubble" differ from the terms "plume" and "vortex"). Looking forward, do you have any recommendations for a general scheme as to how these events can be categorized, in order to separate them from stratospheric smoke plumes that do not show a dynamical signature? Allen et al. (2020) coined a new term for this phenomenon, "Smoke With Induced Rotation and Lofting (SWIRL)", an acronym that accounts for the aerosol source as well as for two of the obvious dynamical aspects of the phenomenon.

**Technical Corrections**

Line 5: change "find" to "finds"

Line 15: change "monoxyde" to "monoxide"

Line 23: change "wildfire" to "wildfires"

Line 43: may want to spell out "CALIPSO"

Line 152: change "event" to "events"

Line 160: Should "Fig. 3a" be "Fig. 3b"?

Line 175: Typo "Fig. fig:CALIOPa"

Line 196: Figure 7 is referred to before Figure 6. I would suggest reversing the order of the figures.

Line 189: change "WaS" to "was"

Line 191: may want to spell out "GOME"

Line 218: change "stations" to "station"

Line 286: What is "beta drift"?

Line 317: change "(c,g)" to "(c,f)"

Line 353: change "wilfires" to "wildfires"

Line 395: change "in a an" to "in an"

Figure 3 Caption: "PVU" is used here, but it isn't defined until the Figure 8 caption. May want to define it here. Also, is the "ERA5 tracking" referred to here using the PV or ozone method?

Figure 5 Caption: I think that "orange" and "green" in the figure caption aren't consistent with the lines on the figure.

Figure A1: What are the axes on these plots?

---

## Referee Comment (RC2) · Anonymous Referee #2 · 13 Dec 2020

*Review on the paper*

**Smoke-charged vortices in the stratosphere generated by wildfires and their behaviour in both hemispheres : comparing Australia 2020 to Canada 2017**

*by Lestrelin, H,. Legras, B. Podglajen, A. and Salihoglu, M.*

The authors re-analysed the evolution of smoke clouds resulting from the wildfires that occurred in Canada in 2017 and Australia in 2019-20. In particular, the authors focus on the PV field (or rather a modified PV field, $\Pi$, referred to as the Lait PV field) together

ozone concentration to track the smoke clouds confined in anticylonic eddies. The use of the PV field provides a natural and pertinent analysis, and is clearly justified in the paper. Overall the paper is well written, interesting and shed new light on the evolution of the events.

**Scientific points**

1. Line 90 Müller & Günther uses $\Pi_g$ for $\epsilon = -4$ and $\Pi_L$ for $\epsilon = 9/2$. Maybe the authors could use the same convention, and add a comment to explain why they use different values of $\epsilon$.

2. Line 114, the authors state 'mean at the same latitude and altitude'. Do they mean a zonal average or a time average?

3. Can the volume integrated PV be determined for each vortex from the available data? If yes, can anything meaningful be discussed, in particular during the vortex evolution and the splitting events? Alternatively, does the nature of the way PV is obtained make such an analysis irrelevant?

**Minor wording points**

1. Line 30, sentence "It is a natural..". Possibly rephrase to read 'Investigating... is a natural extension to [ADD REF(S)].

2. Line 37, maybe insert 'Australian' between '2020' and 'case'

3. Line 116, if the steps $n-1$, $n$ and $n+1$ refer to times, it may be worth mentioning is explicitly.

4. Line 133, 'to dissociate/dissociating': the verb/term 'to split/splitting' is the most often used when discussing vortex breaking.
5. Line 147, Please check the use of the word 'thalweg'.

6. Lin 153, insert 'a' between 'month' and 'half'

7. Line 175, fix the reference to the figure

8. Line 201 'formation' may be better than 'birth'; 'decay' or destruction' may be better than 'loss' (also line 118)

9. Line 279, NH is not explicitly defined. Although line 331 suggests the authors refersto Northern Hemisphere.

10. Line 313, SW is not explicitly defined.

11. Overall revise the punctuation. Some sentences are long and could be split into several shorter sentences. Additional commas could also help readability.

---

## Referee Comment (RC3) · Anonymous Referee #3 · 27 Dec 2020

**General:**
This is an important paper that should be published by ACP after taking into account few points listed below

**Major comments:**

- My strongest criticism, is related to the explanation how the reanalysis data, like

[Figure]

ERA5 does work (sections 2.2.1 and 2.2.2). A more careful explanations would help to understand better this paper, especially if you assume that not every reader is an expert in the assimilation procedure. Because either ECMWF operational analysis nor the ERA5 reanalysis does assimilate the aerosol observations (the only pure observational evidence from CALIOP) it is difficult to imagine that ECMWF/EAR5 data does contain any smoke-related information at all. However, you show that in the PV/ozone fields (Figure 3/7) there are clear signatures of such smoke clouds. Thus, if these structures are reproduced by the reanalysis, the respective assimilation increments should be small...?

- On the other hand, you also show that the assimilation increments within such structures (Figure 9) are really large. Is it true only within such "undetected clouds"? Maybe a separate figure (like Figure 7) but only for the assimilation increments would also help to follow the cloud? In any case I would recommend to explain better the applied method, especially the apparent contradiction between the "resolved" clouds in ERA5 data and unresolved properties manifesting in the "large" assimilation increments.

**Minor comments:**

- L103-106
  difficult to understand...please reformulate (see my main point)

- Figure 10
  You mentioned in section 2.2.1 that you do not use the ERA5 PV but calculate your own PV from eq. (1). How do you proceed for the assimilation increments of PV discussed in section 4.2.2

---

## Short Comment (SC1) · 18 Jan 2021

This is an excellent paper, I enjoyed to read it. However, it can still be improved!

The improvement deals with a better consideration of recent literature on stratospheric smoke observations. The respective lack motivated me (as lidar specialist) to write this small review letter. I hope you enjoy my suggestions!

Please check Baars et al., ACP, 2019!

[Figure]

The unprecedented 2017–2018 stratospheric smoke event: decay phase and aerosol properties observed with the EARLINET Holger Baars, Albert Ansmann, Kevin Ohneiser, Moritz Haarig, Ronny Engelmann, Dietrich Althausen, Ingrid Hanssen, Michael Gausa, Aleksander Pietruczuk, Artur Szkop, Iwona S. Stachlewska, Dongxiang Wang, Jens Reichardt, Annett Skupin, Ina Mattis, Thomas Trickl, Hannes Vogelmann, Francisco Navas-Guzmán, Alexander Haefele, Karen Acheson, Albert A. Ruth, Boyan Tatarov, Detlef Müller, Qiaoyun Hu, Thierry Podvin, Philippe Goloub, Igor Veselovskii, Christophe Pietras, Martial Haeffelin, Patrick Fréville, Michaël Sicard, Adolfo Comerón, Alfonso Javier Fernández García, Francisco Molero Menéndez, Carmen Córdoba-Jabonero, Juan Luis Guerrero-Rascado, Lucas Alados-Arboledas, Daniele Bortoli, Maria João Costa, Davide Dionisi, Gian Luigi Liberti, Xuan Wang, Alessia Sannino, Nikolaos Papagiannopoulos, Antonella Boselli, Lucia Mona, Giuseppe D'Amico, Salvatore Romano, Maria Rita Perrone, Livio Belegante, Doina Nicolae, Ivan Grigorov, Anna Gialitaki, Vassilis Amiridis, Ourania Soupiona, Alexandros Papayannis, Rodanthi-Elisaveth Mamouri, Argyro Nisantzi, Birgit Heese, Julian Hofer, Yoav Y. Schechner, Ulla Wandinger, and Gelsomina Pappalardo Atmos. Chem. Phys., 19, 15183–15198, https://doi.org/10.5194/acp-19-15183-2019, 2019.

These authors show an overview of European lidar network observations of the smoke in 2017. They discuss different lofting possibilities.... They found many apparently ascending structures. Maybe part of these ascending structures can now be explained by the development of smoke-charged vorticities. Please check, and discuss...!

Next paper....

Hu, Q., Goloub, P., Veselovskii, I., Bravo-Aranda, J.-A., Popovici, I. E., Podvin, T., Haeffelin, M., Lopatin, A., Dubovik, O., Pietras, C., Huang, X., Torres, B., and Chen, C.: Long-range-transported Canadian smoke plumes in the lower stratosphere over northern France, Atmos. Chem. Phys., 19, 1173–1193, https://doi.org/10.5194/acp-19-1173-2019, 2019.

[Figure]

This is a nice paper from the very tough French lidar group in Lille. They describe the beginning of the smoke evolution in western Canada in August 2017 in large detail and show measurements over northern France. Should be mentioned (referenced) in the introduction.

Regarding the introduction of your paper:

Page 1-2. Introduction section: The reader, not too familiar with the topic, may get the impression that the smoke self-lifting aspect was introduced by Khaykin et al, 2018, 2020. Yes these authors did a great job concerning the discussion on lofting of smoke. But the basic papers that triggered all this are from Boers et al (2010) and de Laat et al. (2012). You may adjust your introduction... and state that.

Boers, R., de Laat, A. T., Stein Zweers, D. C., and Dirksen, R. J.: Lifting potential of solar-heated aerosol layers, Geophys. Res. Lett., 37, L24802, doi:10.1029/2010GL045171, 2010.

de Laat, A. T. J., Stein Zweers, D. C., Boers, R., and Tuinder, O. N. E.: A solar escalator: Observational evidence of the self-lifting of smoke and aerosols by absorption of solar radiation in the February 2009 Australian Black Saturday plume, J. Geophys. Res., 117, D04204, doi:10.1029/2011JD017016, 2012. And there is this paper of Torres et al, JGR, 2020 .

And in the recent paper of Torres et al. (2020) there is strong focus on lofting of smoke as well.

Torres, O., Bhartia, P. K., Taha, G., Jethva, H., Das, S., Colarco, P., Krotkov, N., Omar, A., and Ahn, C.: Stratospheric Injection of Massive Smoke Plume from Canadian Boreal Fires in 2017 as seen by DSCOVR‐EPIC, CALIOP and OMPS‐LP Observations, Journal of Geophysical Research: Atmospheres, 125, e2020JD032579, https://doi.org/10.1029/2020JD032579, 2020a.

That paper should also be mentioned, I think!

Finally, we (Ohneiser, Ansmann et al. . .) published the first (!) paper on the Australian 2020 smoke. You are probably not aware of it. No problem! And we already needed the self-lifting effect to find agreement with the HYSPLIT trajectories and our lidar observations. And we saw the bubble vortex KOOBOR over Punta Arenas, Chile, from 20-27 January 2020, the lifting rate was about 1 km per day, ... all this can be found in that paper. The paper should also be mentioned in your article.

Ohneiser, K., Ansmann, A., Baars, H., Seifert, P., Barja, B., Jimenez, C., Radenz, M., Teisseire, A., Floutsi, A., Haarig, M., Foth, A., Chudnovsky, A., Engelmann, R., Zamorano, F., B\"uhl, J., and Wandinger, U.: Smoke of extreme Australian bushfires observed in the stratosphere over Punta Arenas, Chile, in January 2020: optical thickness, lidar ratios, and depolarization ratio sat 355 and 532∼nm, Atmos. Chem. Phys., 20, 8003-8015, https://doi.org/10.5194/acp-20-8003-2020, 2020.

Last question: Why did the 2017 smoke ended up at about 23 km? Baars et al. (2019) argued ...because of the Brewer Dobson circulation and the phase of QBO to that time (August-October 2017)? Is that the right answer?
* * *

---

## Author Comment (AC1) · 2 Mar 2021

[acpd,discussion]copernicus

bernard.legras@lmd.ipsl.fr Lestrelin, Legras, Podglajen, Salihoglu

[Figure]

**Answer to reviewer #2**

Scientific points

1. *Line 90 Müller & Günther uses $\Pi_g$ for $\epsilon = -4$ and $\Pi_L$ for $\epsilon = 9/2$. Maybe the authors could use the same convention, and add a comment to explain why they use different values of $\epsilon$.*

   We have added a comment regarding the choice of $\epsilon$, which depends on the background temperature profile. However, we prefer to leave the notation as is to avoid introducing unnecessary new symbols, since the precise value of $\epsilon$ is not central to our argumentation and mainly included so that the analysis is reproducible.

2. *Line 114, the authors state 'mean at the same latitude and altitude'. Do they mean a zonal average or a time average?*

   It is a zonal mean.

3. *Can the volume integrated PV be determined for each vortex from the available data? If yes, can anything meaningful be discussed, in particular during the vortex evolution and the splitting events? Alternatively, does the nature of the way PV is obtained make such an analysis irrelevant?*

   This is an interesting diagnostic that we have not yet developed and which could be combined with the aerosol sections provided by CALIOP. We would like to delay this development to another work in which we will study the life cycle of the vortices.

Minor wording points

1. *Line 30, sentence 'It is a natural..'. Possibly rephrase to read 'Investigating... is a*

*natural extension to [ADD REF(S)]*

Done

2. *Line 37, maybe insert 'Australian' between '2020' and 'case'*

Done

3. *Line 116, if the steps n−1, n and n+ 1 refer to times, it may be worth mentioning is explicitly.*

Done

4. *Line 133, 'to dissociate/dissociating': the verb/term 'to split/splitting' is the most often used when discussing vortex breaking.*

Agreed and corrected

5. *Line 147, Please check the use of the word 'thalweg'.*

It has been replaced by the more common "trough".

6. *Lin 153, insert 'a' between 'month' and 'half'.*

Done

7. *Line 175, fix the reference to the figure*

Fixed

8. *Line 201 'formation' may be better than 'birth'; 'decay' or destruction' may be better than 'loss' (also line 118)*

Birth has been changed into formation but loss has been kept as we cannot say how long the vortices survived after we could not track them anymore. We have added a sentence mentioning this point.

9. *Line 279, NH is not explicitly defined. Although line 331 suggests the authors refer to Northern Hemisphere.*

   The few instances of NH and SH have been expanded.

10. *Line 313, SW is not explicitly defined.*

    Expanded to shortwave

11. *Overall revise the punctuation. Some sentences are long and could be split into several shorter sentences. Additional commas could also help readability.*

    We tried, however, to do our best in this respect. We have cut a few long sentences and added some commas.

---

## Author Comment (AC2) · 2 Mar 2021

[acpd,discussion]copernicus

bernard.legras@lmd.ipsl.fr Lestrelin, Legras, Podglajen, Salihoglu

[Figure]

**Answer to reviewer #1**

General comments

*This paper details the dynamics of a newly-discovered phenomenon associated with large pyroCb smoke plumes in the stratosphere, namely the self-organized anticyclonic vortices that are formed due to absorption of solar radiation by black carbon within the plumes. A comparison is made between the 2019-2020 Australian plumes and several plumes associated with large Canadian fires in 2017. Detailed analysis of the Canadian plumes use the Lait Potential Vorticity (PV) from ERA5 to track the vortex evolution and to examine the composite dynamical structures of the vortex. ERA5 ozone is also used to successfully track the vortex locations. Composites of PV, temperature, and ozone help to further understand these features. The paper also details how these features are maintained in the analyses by assimilation of temperature and ozone data, and it examines various dynamical indices to test whether the vortices are in balance and/or inertially unstable. This paper provides an excellent addition to the study of smoke-induced dynamics.*

We appreciate the evaluation of the reviewer and his numerous detailed comments.

Specific comments

- *Line 10: You use the term "low absolute potential vorticity" here. Just to be clear, does this mean low magnitude (i.e., absolute value) of the potential vorticity?*

  We mean small absolute vorticity and small potential vorticity as well.

- *Line 83: I don't think "g is the free-fall acceleration" is necessary, since "g" isn't in Eq. (1).*

We had omitted the required constant $-g$ factor in the definition. It is now restored.

- *Lines 112-114: Quoting from the paper, "we used both $\Pi$ and the ozone anomaly defined as the deviation with respect to the mean at the same latitude and altitude." So are both $\Pi$ and ozone defined using the anomaly with respect to the zonal mean, or is just ozone calculated as the anomaly, while $\Pi$ is the raw value from Eq. (2)? This sentence could be read either way.*

Only ozone is defined using the anomaly with respect to the zonal mean. The sentence has been modified.

- *Lines 112-114: As mentioned by the authors, the commonly used PV has a disadvantage of large background vertical gradient. While Khaykin et al. (2020) used relative vorticity, Kablick et al. (2020) used the "regular" PV to analyze the 2019-2020 Australian smoke plumes. They used the PV anomaly relative to the zonal mean in units of percent of the absolute value of the zonal mean PV. This alternate approach also reduces the influence of the large background gradient.*

We agree there are several ways in order to remove the background gradient in the display of the vertical structure. The Lait PV has the advantage to remain an adiabatic invariant of motion.

- *Line 139-142: You talk about the "kernel of almost zero PV and low ozone...." To visualize this statement, it would be interesting to see latitude/altitude cross-sections of PV and ozone along the CALIPSO track to compare with Figures 1 and 2.*

The latitude/altitude and longitude/altitude sections are displayed in the animation of vortex O in the supplement and are indeed very instructive. They are centered on the IFS vortex to provide continuity. Doing the same with vortices A, B1 and B2 brings less additional information. Figure 1 and 2 show the location of the vortex

center according to the PV and ozone tracking. As these centers are mimima in the 3D domain, surrounding contours can be plotted but this would totally obscure the aerosol pattern on many panels.

- *Line 147 (also Lines 158 and 394) : I am unfamiliar with the term "thalweg". Are you referring to a trough? Could you provide some dynamical field on the maps to indicate where this is occurring to help visualize the point you're making?*

  We mean trough and this has been corrected. Montgomery potential contours have been added to all the isentropic charts of the animations.

- *Line 194-195: Is the statement "it differs from the 2020 case where such effect is not observed for any of the three vortices" referring to results in Khaykin et al. (2020)?*

  Yes. The sentence has been improved.

- *Section 3.5. This section provides a helpful reference of other papers that have studied this event. The PV anomaly associated with smoke during the Canadian event was also examined in a recent paper by Allen et al. (2020). While that paper focused primarily on the "Koobor" vortex, they also included a PV anomaly map for 28 August 2017 associated with what this paper calls "Vortex A" (see Figure 16 of the following reference). Allen, D. R., M. D. Fromm, G. P. Kablick III, and G. E. Nedoluha, 2020: Smoke With Induced Rotation and Lofting (SWIRL) in the Stratosphere, J. Atmos. Sci., 77, 4297-4316, https://doi.org/10.1175/JAS-D-20-0131.1.*

  We missed this paper which was published at a time close to our submission. It is now referred and commented. Indeed the structure shown in the figure 16 of this work displays the vortex A on 28 August but the structure provided by MERRA-2 is apparently much less compact than in the ERA5, perhaps due to a lower spatial resolution in the vertical and horizontal directions. The elongated vortex O is also visible in several panels of this figure.

- *Line 243-245: It is interesting that the 2017 case doesn't show the temperature dipole. Is that simply due to the contours chosen (i.e., the warm anomaly is really there, but it is less than 1.0 K)?*

  There is indeed a warm anomaly of weaker amplitude which is actually responsible for the cooling pattern. A new version of the figure which shows the warm anomaly contours is now provided.

- *Line 252: It is unclear exactly how you calculate the horizontal length scale $L_h$. The text says it is defined as "the diameter of the ring of wind speed maximum". Do you calculate the diameter from the wind speed explicitly for each case? If so, does the wind speed calculation involve removing the background wind in order to focus on the wind associated with the anticyclone?*

  We estimate the length explicitly for each case along the East-West direction (i.e. between local extrema of the meridional velocity). The background wind is not removed in the calculation, since by definition its variations at the scale of the vortex are small so that they do not affect local extrema.

- *Line 256: Similarly, $L_z$ is "the vertical extent of the vorticity contour at maximum wind speed". Is this calculated explicitly? Would it be possible to include some more details on this calculation?*

  Yes. This is now explained with more details.

- *Line 257: You say "the 2017 vortex A being about 8 times smaller in volume than its gigantic 2020 counterpart". This difference seems too large. If the cloud is considered as a cylinder, then volume is $V = \pi(L_h/2)^2 L_z$. Using numbers from Table 1 we get $V_A = \pi(686/2)^2 3.5 = 1.3 \times 10^6 \text{km}^3$, and $V_{Koobor} = \pi(784/2)^2 6.1 = 2.9 \times 10^6 \text{km}^3$. So Koobor is only 2.9/1.3=2.2 times larger in volume than vortex A. Are these estimates correct, or do you use another method to estimate volume?*

You are correct, thank you for pointing this out. We had previously estimated the volume of the vortices using the same value of vorticity threshold for both cases, but it is more consistent to adapt the value to each case. This has now been corrected.

- *Line 263: You may want to define the condition for inertial instability here.*

The criterion of vanishing PV is actually formally related to symmetric instability (Hoskins, 1974) but, here, we see that the absolute vorticity (not shown) also vanishes, therefore the symmetric instability is an inertial instability.

- *Line 328-330: It looks like the dates used for the Australian vortex are 2-27 February in Figure 8 (mean structure), Figure 9 (heating rate, temperature and vorticity tendencies), and Figure 11 (ozone tendencies), but for Figure 10 (PV increments) the dates are 7-19 January. It there a particular reason that different dates are chosen for PV? Are the mean composites and increments of PV much different if you calculate them for the different periods? Also, are the green lines on Figure 10 from 7-19 January or 2-27 February?*

There was a mistake here, thank you for spotting it. We have replaced the figure in the main body of the paper by the correct time period, while the 7-19 January period figure has been moved to the appendix. Note that the green contours always correspond to the vorticity in the time period indicated.

The reason for showing the January periods is to emphasize that the LPV increment dipole does not tilt along the direction of the background wind (which corresponds to the drift of the vortex) but rather along the direction of the wind shear, a property which might be found counter intuitive. We follow the reviewers advice and now show this feature in the appendix.

The composite anomaly and increment structures are qualitatively consistent between different periods, but there can be quantitative differences, some of those

related to the composite are reported in Tab. 1 of the paper. Regarding the incre-
ments, there are differences associated for instance with changes in the ascent
rate of the vortex in altitude coordinate which is larger, for Koobor, in January
(250 m/day) than in February (150 m/day). This is mentioned in the revised pa-
per together with the new figure in the appendix. The choice of the February time
period fr the main body of the paper is motivated by the relative steadiness of the
structure during that month (see Fig. 7b of Khaykin et al. (2020), which shows a
rather constant relative vorticity maximum).

- *Line 335: The northwest-southeast tilt of the PV increments for the Koobor vortex,*
  *shown in Figure 10a, is interesting. In the recent analysis by Allen et al. (2020),*
  *they examined how Koobor tilts with height and found a NW-SE tilt of the vortex*
  *in January. They used a dynamical argument to show how this tilt may develop*
  *from internal vortex dynamics in a shear flow. The PV increments shown in this*
  *paper appear to support this observed dynamical structure. Also, would the same*
  *argument you make in Appendix B apply to the ozone structure seen in Figure*
  *11?*

The argument of Allen et al. (2020) concerns the structure of the vortex and
does not apply to the increments, although they are both related to the vertical
shear. We now mention their explanation. Vacillations of vortices in a shear flow
constitute a fairly standard topic which has been studied a lot, included by the
authors in 2D (Legras et al., 2001) and, in 3D, by David Dritschel and collabora-
tors (e.g., Tsang and Dritschel, 2015). There are also a large number of works
in the oceanographic literature. Therefore Koobor and its siblings were expected
to follow the general laws and to undergo vacillation, erosion and splitting when
submitted to a shear. Such vacillations are obvious in Fig. 6b of Khaykin et al.
(2020). Koobor went through a first splitting by 25 February 2020, which was
actually well predicted by the ECMWF opertaional model a few days in advance,
and a second one by 30 March. For the Canadian case, the deformation of vortex

O in the trough led to its splitting in at least three parts that we followed.

Our reasoning in appendix B applies to the ozone increment (or the increment or any quantity organized as a monopole).

- *Line 344: Does "low absolute PV" mean low magnitude (i.e., low absolute value of PV)?*

Yes. The model provides PV and absolute vorticity values close to 0 or even slightly negative. The contrast is such with the ambient PV that the practical situation is that of a ball of zero PV immersed into the stratosphere. Such contrast is made possible by the vertical motion. The familiar formation of cut-off lows or cut-off highs by horizontal isentropic motion never reaches such a situation.

- *Line 408: May want to define terms explicitly in the text here, particularly $W$ and $\Lambda$. I assume these are vertical wind and vertical shear of the zonal wind. They are indirectly defined in the Table B1, but not in the text.*

Done.

- *Line 416: How is the wind shear estimate calculated here (i.e., what time range is used)?*

The time range of the composite, this is now specified.

- *Figure 3 caption: Could you include in the caption what time of day was used for the PV analyses?*

Done

- *Figure S4: Is this calculated with normal PV or with the Lait PV? Also, as a reference, it would be useful to include the zonal mean PV on this plot. This should become increasingly negative with time as the parcel ascends. Do the Canadian plumes show a similar behavior?*

This is the standard PV. The anomaly is defined with respect to the zonal mean PV at same altitude. Therefore this later is the difference between the blue and the orange curves. The initial stage of vortex O exhibits a similar behaviour in 2017. After its split in 3 parts, the offspring exhibit non zero absolute vorticity at the core. As such events usually generate only a small amount of in-mixing, it is unclear whether the effect is real or due to the limitation of assimilation.

- *I enjoyed the animations in the supplementary material. I assume the PV used in the animations is the normal PV, not the Lait PV, right?*

  Yes

- *Lastly, there are quite a few different names used in this paper for different aspects of this new phenomenon. For example the terms "smoke-charged vortex", "smoke charged pancake vortex", "smoke vortex", "smoke plume", "smoke bubble", and just "bubble" could possibly be condensed into fewer descriptions. The Australian plumes are called "Koobor", "2nd Vortex", and "3rd Vortex", while the Canadian plumes are "Vortex O" (also called "mother vortex"), "Vortex A", "Vortex B1", and "Vortex B2". Different terms are also used for "Koobor", such as "main vortex", and "major vortex". Given this is such a new discovery, to avoid potential confusion, terms could be consolidated and defined (e.g., how does the term "bubble" differ from the terms "plume" and "vortex"). Looking forward, do you have any recommendations for a general scheme as to how these events can be categorized, in order to separate them from stratospheric smoke plumes that do not show a dynamical signature? Allen et al. (2020) coined a new term for this phenomenon, "Smoke With Induced Rotation and Lofting (SWIRL)", an acronym that accounts for the aerosol source as well as for two of the obvious dynamical aspects of the phenomenon.*

  We agree that too much gratuitous variations is undesirable and should be blamed. We have reduced them in the revised version. However, we still find

useful to have different words to name the smoke bubble or the unshaped smoke plume as seen purely by CALIOP and to name the vortex identified from the reanalysis. We are not very fond of SWIRL as it suggests that heating can generate by itself mean PV or rotation, which is wrong. Swirl is also used already in the fluid dynamics literature, often associated with the word vortex (see, e.g. Mitrofanova et al., 2013), which might be confusing, and has unpleasant meanings in slang. We offer our own acronym, ASTuS (Ascending Smoke Turbo in the Stratosphere) where turbo is a latin word for vortex, which does not have these drawbacks.

Noticeable technical corrections

- *Line 286: What is "beta drift"?*

  The beta-drift is a old concept of vortex dynamics on a rotating planet which is standard in the literature on tropical cyclones (see, e.g., Wang and Li, 1992). The beta-drift occurs for a vortex immersed within a planetary PV gradient and results, for a cyclone, into a poleward and westward motion with respect to the mean flow. This is changed to an equatorward motion for an anticyclone. The ingredient of the beta-drift is the wrapping of planetary vorticity around the vortex (Sai-Lap Lam and Dritschel, 2001). This effect is visible in Fig.10 of Allen et al. (2020) but the beta-drift is ignored by these authors who privilege the effect of the vertical tilt.

- *Figure 5 Caption: I think that "orange" and "green" in the figure caption aren't consistent with the lines on the figure.*

  There are actually two curves on each panel of the figure but because the PV tracking and the ozone tracking are very close, they are most often superimposed.

- All other corrections have been applied
* * *
Interactive comment

**References**

Allen, D. R., Fromm, M. D., Kablick III, G. P., and Nedoluha, G. E.: Smoke with Induced Rotation and Lofting (SWIRL) in the Stratosphere, Journal of the Atmospheric Sciences, 77, 4297–4316, https://doi.org/10.1175/JAS-D-20-0131.1, 2020.

Hoskins, B. J.: The Role of Potential Vorticity in Symmetric Stability and Instability, Quarterly Journal of the Royal Meteorological Society, 100, 480–482, https://doi.org/10.1002/qj.49710042520, 1974.

Khaykin, S., Legras, B., Bucci, S., Sellitto, P., Isaksen, L., Tence, F., Bekki, S., Bourassa, A., Rieger, L., Zawada, D., Jumelet, J., and Godin-Beekmann, S.: The 2019-2020 Australian wildfires generated a persistent smoke-charged vortex rising up to 35 km altitude, Communications Earth and Environment, 1, 22, https://doi.org/10.1038/s43247-020-00022-5, 2020.

Legras, B., Dritschel, D. G., and Caillol, P.: The Erosion of a Distributed Two-Dimensional Vortex in a Background Straining Flow, Journal of Fluid Mechanics, 441, 1–16, https://doi.org/10.1017/S002211200100502X, 2001.

Mitrofanova, O. V., Podzorov, G. D., and Pozdeeva, I. G.: Vortex Structure of Swirl Flows, International Journal of Heat and Mass Transfer, 65, 225–234, https://doi.org/10.1016/j.ijheatmasstransfer.2013.06.007, 2013.

Sai-Lap Lam, J. and Dritschel, D. G.: On the Beta-Drift of an Initially Circular Vortex Patch, Journal of Fluid Mechanics, 436, 107–129, https://doi.org/10.1017/S0022112001003974, 2001.

Tsang, Y.-K. and Dritschel, D. G.: Ellipsoidal Vortices in Rotating Stratified Fluids: Beyond the Quasi-Geostrophic Approximation, Journal of Fluid Mechanics, 762, 196–231, https://doi.org/10.1017/jfm.2014.630, 2015.

Wang, B. and Li, X.: The Beta Drift of Three-Dimensional Vortices: A Numerical Study, Monthly Weather Review, 120, 579–593, https://doi.org/10.1175/1520-0493(1992)120<0579:TBDOTD>2.0.CO;2, 1992.

---

## Author Comment (AC3) · 2 Mar 2021

[acpd,discussion]copernicus

bernard.legras@lmd.ipsl.fr Lestrelin, Legras, Podglajen, Salihoglu

**Answer to reviewer #3**

Major comment

*My strongest criticism, is related to the explanation how the reanalysis data, like ERA5 does work (sections 2.2.1 and 2.2.2). A more careful explanations would help to understand better this paper, especially if you assume that not every reader is an expert in the assimilation procedure. Because either ECMWF operational analysis nor the ERA5 reanalysis does assimilate the aerosol observations (the only pure observational evidence from CALIOP) it is difficult to imagine that ECMWF/EAR5 data does contain any smoke-related information at all. However, you show that in the PV/ozone fields (Figure 3/7) there are clear signatures of such smoke clouds. Thus, if these structures are reproduced by the reanalysis, the respective assimilation increments should be small...?*

*On the other hand, you also show that the assimilation increments within such structures (Figure 9) are really large. Is it true only within such "undetected clouds"? Maybe a separate figure (like Figure 7) but only for the assimilation increments would also help to follow the cloud? In any case I would recommend to explain better the applied method, especially the apparent contradiction between the "resolved" clouds in ERA5 data and unresolved properties manifesting in the "large" assimilation increments.*

It is clearly beyond the scope of our work to provide a tutorial on assimilation which is a whole field by itself but we have tried to add some sentences to help the reader who is not familiar. And yes it is a wonder that the model reproduces a smoke vortex even if it does not contain smoke. The "miracle" is due to the fact that the vortex exhibits a strong thermal signature which is well detected by the satellites. This pattern is forced into the model by the assimilation and through the principle that the motion is balanced (McIntyre, 2015), the whole vortex is reconstructed with some accuracy. Over the continental regions the assimilation also uses ground based radio-soundings that contain

wind information. Therefore there is no contradiction between resolved and unresolved structures. The missing smoke heating rate is replaced by the assimilation increment, hence the amplitude of this later. In the supplement of Khaykin et al. (2020), it was shown how the deviation of the observations, with respect to the a priori simulated by the model, is reduced by the assimilation, and therefore how the information is used to guide the model. In this study, we are looking at the assimilation from the point of view of the model world and we diagnose the effect of the assimilation on the temperature and PV field. The delicate point is that the temperature assimilation increment cannot be simply interpreted as a heating rate because it diagnoses the final equilibrated state where momentum and temperature are in balance. Assimilation is an iterative procedure where the model is disturbed to get closer to the observation but it is done in such a way that the modification does not add transient riddles of gravity waves. It was shown in Khaykin et al. (2020), that if suddenly the assimilation procedure of new observations is stopped, which is what is done to produce a weather forecast, the vortex does not rise anymore but stays on the same isentropic level and its amplitude decays in about one week. We show in this study that this decay is mostly due to the longwave radiative exchanges that damp the thermal dipolar structure. This is done mostly by carbon dioxide and water vapour. In the real vortex, the longwave radiative effect of the aerosols can only accelerate this damping.

Minor comment

- *L103-106 difficult to understand...please reformulate (see my main point)*

  Several additions have been made to this section to improve clarity.

- *Figure 10 You mentioned in section 2.2.1 that you do not use the ERA5 PV but calculate your own PV from eq. (1). How do you proceed for the assimilation increments of PV discussed in section 4.2.2*

  We use our own calculation to get PV at the full vertical resolution of the model

and not on a selected number of levels as provided by ECMWF. PV is calculated in the same way for the a priori state resulting from the forecast and for the new analysis. The increment is defined as the difference between these two quantities exactly as for the temperature.

**References**

Khaykin, S., Legras, B., Bucci, S., Sellitto, P., Isaksen, L., Tence, F., Bekki, S., Bourassa, A., Rieger, L., Zawada, D., Jumelet, J., and Godin-Beekmann, S.: The 2019-2020 Australian wildfires generated a persistent smoke-charged vortex rising up to 35 km altitude, Communications Earth and Environment, 1, 22, https://doi.org/10.1038/s43247-020-00022-5, 2020.

McIntyre, M.: DYNAMICAL METEOROLOGY | Balanced Flow, in: Encyclopedia of Atmospheric Sciences, pp. 298–303, Elsevier, https://doi.org/10.1016/B978-0-12-382225-3.00484-9, 2015.

---

## Author Comment (AC4) · 2 Mar 2021

[acpd,discussion]copernicus

bernard.legras@lmd.ipsl.fr Lestrelin, Legras, Podglajen, Salihoglu

[Figure]

**Answer to Albert Ansmann**

We thank Albert Ansmann for his appraisal of our work and for providing a complementary list of references. We were actually aware of these references but one. They were not included as our purpose was not to review the whole literature on smoke clouds, in particular from ground lidars, and the works that focus on the determination of the optical properties or the mass distribution of aerosols, two topics that are outside of our work. We quoted Ansmann et al. (2018), as this study shows results directly related to our work. We have, however decided to quote Torres et al. (2020) and Baars et al. (2019), as they both contain informations that we can comment from our work. In particular Torres et al. (2020) provides a detailed description of the early stage based on CALIOP and the images of the EPIC camera at Lagrange point. We also quote de Laat et al. (2012) for its precursor ideas. Boers et al. (2010) is only concerned by smoke patches in the troposphere, Hu et al. (2019) is almost entirely dedicated to the optical properties of the aerosols and we are not discussing any observational results on the 2020 Australian case which is the focus of Ohneiser et al. (2020).

We stress that none of these previous works recognizes the importance of confinement by the vortical structure generated by the immersion of low PV in the stratosphere, the dipole thermal structure of the resulting equilibrated anomaly, the strong induced long wave relaxation (which is not a cooling) and, in general, the implications of an equilibrated response to heating.

We have also concerns about a number of large estimates of rising/heating rates that we do not recover in Khaykin et al. (2020) or in our work. It is easy to generate overestimates of rising rates from a structure moving over a fixed observatory. For instance, Ohneiser et al. (2020), and yourself in your comment claim an ascent rate of 1km/day when Koobor was near the tip of South America which is about three time the ascent rate found by Khaykin et al. (2020) (corroborated by Allen et al. (2020)) at the same time on the trajectory. The ascent of Koobor was basically smooth for three months, especially in potential temperature, except during two episodes of vertical split and it rose from about 16 km to 35 km, which is a considerable way against the Brewer-Dobson circulation but it did not reach the ionosphere. Our view is that only careful Lagrangian tracking of the structures can reliably assess the ascent rate. As we intent to develop this point in a future work, and review the existing literature in this respect, we do not wish to include an incomplete discussion in the present work.

Baars et al. (2019) show interesting evidence of aerosol patches over Europe and the Mediterranean area reaching 23 km by mid September and again in mid December. Although unstated, they do not expect the second patches to be the remnant of the first due to the mean descending circulation. They invoke a circuit identified by Kloss et al. (2019), where smoke patches were injected into the tropics by early September and rose then slowly to higher levels and came back to the mid latitudes carried by the Brewer-Dobson meridional circulation. There is, however, a hole in Baars et al. (2019) reasoning which is that the tropical rise is reported to reach 21 km by March 2018 while the aerosols supposedly blown away by the Brewer-Dobson are found at 23 km over Europe in December 2017. It is now clear that the missing piece of the puzzle is provided by vortex A which reached 23 km by late September leaving a tail along its path.

As for the culmination of the smoke at 23 km in 2017, we see that the center of the vortex A reached 23 km at the end of our tracking which means that the top was at 24-25 km. Besides this we cannot say that the vortex disappeared and stopped rising after we lost its track with the ERA5 potential vorticity. **?** mention that the return to the extratropics was caped at 23 km by the properties of the Brewer-Dobson circulation. This hypothesis is plausible but is not demonstrated due to the fact that the fast ascent of vortex A also culminated at 23 km.

**References**

Allen, D. R., Fromm, M. D., Kablick III, G. P., and Nedoluha, G. E.: Smoke with Induced Rotation and Lofting (SWIRL) in the Stratosphere, Journal of the Atmospheric Sciences, 77, 4297–4316, https://doi.org/10.1175/JAS-D-20-0131.1, 2020.

Ansmann, A., Baars, H., Chudnovsky, A., Mattis, I., Veselovskii, I., Haarig, M., Seifert, P., Engelmann, R., and Wandinger, U.: Extreme Levels of Canadian Wildfire Smoke in the Stratosphere over Central Europe on 21–22 August 2017, Atmospheric Chemistry and Physics, 18, 11 831–11 845, https://doi.org/10.5194/acp-18-11831-2018, 2018.

Baars, H., Ansmann, A., Ohneiser, K., Haarig, M., Engelmann, R., Althausen, D., Hanssen, I., Gausa, M., Pietruczuk, A., Szkop, A., Stachlewska, I. S., Wang, D., Reichardt, J., Skupin, A., Mattis, I., Trickl, T., Vogelmann, H., Navas-Guzmán, F., Haefele, A., Acheson, K., Ruth, A. A., Tatarov, B., Müller, D., Hu, Q., Podvin, T., Goloub, P., Veselovskii, I., Pietras, C., Haeffelin, M., Fréville, P., Sicard, M., Comerón, A., Fernández García, A. J., Molero Menéndez, F., Córdoba-Jabonero, C., Guerrero-Rascado, J. L., Alados-Arboledas, L., Bortoli, D., Costa, M. J., Dionisi, D., Liberti, G. L., Wang, X., Sannino, A., Papagiannopoulos, N., Boselli, A., Mona, L., D'Amico, G., Romano, S., Perrone, M. R., Belegante, L., Nicolae, D., Grigorov, I., Gialitaki, A., Amiridis, V., Soupiona, O., Papayannis, A., Mamouri, R.-E., Nisantzi, A., Heese, B., Hofer, J., Schechner, Y. Y., Wandinger, U., and Pappalardo, G.: The Unprecedented 2017–2018 Stratospheric Smoke Event: Decay Phase and Aerosol Properties Observed with the EARLINET, Atmospheric Chemistry and Physics, 19, 15 183–15 198, https://doi.org/10.5194/acp-19-15183-2019, 2019.

Boers, R., de Laat, A. T., Zweers, D. C. S., and Dirksen, R. J.: Lifting Potential of Solar-Heated Aerosol Layers, Geophysical Research Letters, 37, https://doi.org/10.1029/2010GL045171, 2010.

de Laat, A. T. J., Stein Zweers, D. C., Boers, R., and Tuinder, O. N. E.: A Solar Escalator: Observational Evidence of the Self-Lifting of Smoke and Aerosols by Absorption of Solar Radiation in the February 2009 Australian Black Saturday Plume: OBSERVATIONS OF SELF-LIFTING AEROSOLS, Journal of Geophysical Research: Atmospheres, 117, n/a–n/a, https://doi.org/10.1029/2011JD017016, 2012.

Hu, Q., Goloub, P., Veselovskii, I., Bravo-Aranda, J.-A., Popovici, I. E., Podvin, T., Haeffelin, M., Lopatin, A., Dubovik, O., Pietras, C., Huang, X., Torres, B., and Chen, C.: Long-Range-Transported Canadian Smoke Plumes in the Lower Stratosphere over Northern France, Atmospheric Chemistry and Physics, 19, 1173–1193, https://doi.org/10.5194/acp-19-1173-2019, 2019.

Khaykin, S., Legras, B., Bucci, S., Sellitto, P., Isaksen, L., Tence, F., Bekki, S., Bourassa, A., Rieger, L., Zawada, D., Jumelet, J., and Godin-Beekmann, S.: The 2019-2020 Australian wildfires generated a persistent smoke-charged vortex rising up to 35 km altitude, Communications Earth and Environment, 1, 22, https://doi.org/10.1038/s43247-020-00022-5, 2020.

Kloss, C., Berthet, G., Sellitto, P., Ploeger, F., Bucci, S., Khaykin, S., Jégou, F., Taha, G., Thomason, L. W., Barret, B., Le Flochmoen, E., von Hobe, M., Bossolasco, A., Bègue, N., and Legras, B.: Transport of the 2017 Canadian wildfire plume to the tropics via the Asian monsoon circulation, Atmospheric Chemistry and Physics, 19, 13 547–13 567, https://doi.org/10.5194/acp-19-13547-2019, 2019.

Ohneiser, K., Ansmann, A., Baars, H., Seifert, P., Barja, B., Jimenez, C., Radenz, M., Teisseire, A., Floutsi, A., Haarig, M., Foth, A., Chudnovsky, A., Engelmann, R., Zamorano, F., Bühl, J., and Wandinger, U.: Smoke of Extreme Australian Bushfires Observed in the Stratosphere over Punta Arenas, Chile, in January 2020: Optical Thickness, Lidar Ratios, and Depolarization Ratios at 355 and 532 Nm, Atmospheric Chemistry and Physics, 20, 8003–8015, https://doi.org/10.5194/acp-20-8003-2020, 2020.

Torres, O., Bhartia, P. K., Taha, G., Jethva, H., Das, S., Colarco, P., Krotkov, N., Omar, A., and Ahn, C.: Stratospheric Injection of Massive Smoke Plume From Canadian Boreal Fires in 2017 as Seen by DSCOVR-EPIC, CALIOP, and OMPS-LP Observations, Journal of Geophysical Research: Atmospheres, 125, https://doi.org/10.1029/2020JD032579, 2020.

---

## Author Response (AR2)

Editor Decision: Publish subject to minor revisions (review by editor) (20 Mar 2021) by Peter Haynes
Comments to the Author:
I'd like to accept this without going back to the referees. In looking through the revised version of the paper I noticed some minor technical points which are listed below. Please can you address those.

More substantially, your reply to Referee 3 is not very clear about whether or not you have actually made any changes in response to the request for more information about assimilation etc. Your general point is fine -- that one can't included a tutorial on assimilation and you have given some useful comment in the reply. My impression is that you have made various changes in the text that make some of the important points re assimilation clearer -- but they are little scattered around in the text. Please can you provide an updated reply that makes it clear that you have included a bit more information/clarification on this topic -- and alongside that consider whether the information could be better organised in the text (and include that point in the updated reply if you do change something further).

We believe that the main reservation from referee 3 came from the fact that he (or she) does not see how one can produce a smoke vortex without smoke. We tried to clarify this issue in detail in the answer and we have added a few sentences in section 2.2.2 to explain how wind and vorticity can be retrieved from information mainly based on temperatures. This leads us to the notion of balance that we cannot fully discuss within the scope of this work and we refer to Mc Intyre, 2015, which is, in our opinion, an accessible discussion of this notion. The image below is extracted from the difference file produced by latexdiff between the submitted version and the updated revision. Additions are in blue. We refer also in our answer to the fact that the temperature increment cannot be interpreted as a heating rate. This is implicit in section 2.2.2 and is explained in section 4.2.1. We did not see what to add to this latter. Other points of the answer refer to the previous work of Khaykin et al. (2020). We have added the reference to our two sections in the answer.

**2.2.2 Assimilation increment**

The ERA5 is constrained by observations over repeated 12-hour assimilation cycles.  Over each cycle, the assimilation increment is defined  as the difference between the new analysis and the first guess provided as a final stage of a free forecast run of the model, initialized from the previous analysis 12 hours before. This definition can be applied to any of the basic variables of the model or to derived quantities like potential vorticity. In the ERA5, the assimilation increments can be calculated on each day at 6:00 and 18:00 UTC. In order to diagnose how the observations are forcing the vortices, we calculated the assimilation increments of temperature, vorticity, potential vorticity and ozone. Temperature and ozone determine the radiances that are measured by spaceborne instruments and are also directly accessible from in situ instruments. On the contrary, potential vorticity cannot be directly retrieved from any instrument and is indirectly constrained (see below). These three parameters are updated by the assimilation system in order to reduce the difference between observed quantities (typically radiances but also deviations of the GPS signal path) and simulated quantities (radiances that a satellite flying "above the model" would see). It is tempting to see the temperature assimilation increment as an additional heating but this is incorrect . The increment is calculated from an adjusted state, resulting from the iterations of the assimilation, in which both temperature and motion respond to the forcing by the observations . As wind observations are much sparser than temperature observations, one would expect that analysis winds, and related quantities like

4

potential vorticity, are more poorly constrained and therefore less accurate than analysis temperatures. While this statement is true to a large extent in the tropics, in the mid-latitudes the temperature and wind fields are related through thermal wind balance. This equilibrium is enforced by the assimilation system which filters out the transient modes that deviate from it. Hence, thanks to this miracle of assimilation, assimilating the temperature signal of the vortex is sufficient to reconstruct the whole thermal and dynamical field associated with the balanced structure (McIntyre, 2015).

 It should be noted here that neither the ECMWF operational analysis nor the ERA5 assimilate aerosol observations. The smoke plumes are  totally absent from the IFS, where stratospheric aerosols were only accounted by mean climatological distribution during the periods of investigation, and it is only their dynamical vortical signature which are introduced in the model as described above.
* * *
I then expect to accept the paper (and will be able to assure Referee 3 that the their recommendation has been properly considered and that an appropriate response has been provided).

l10: "We analyze the dynamical structure of the vortices produced by these two wildfires and demonstrate how they are maintained
by the assimilation of data from instruments measuring the signature of the vortices in the temperature and ozone field."

Needs changing -- it is not data assimilation that maintains the vortices -- it is "assimilation of the real temperature and ozone signatures of the vortices that explains the appearance and maintenance of the vortices in the constructed dynamical fields"

The sentence kindly provided by the editor has replaced the previous version. We have tried to improve our english in the revised version, in particular to remove americanisms, but clearly this is not yet fully satisfactory.

l39: "In particular, a stratospheric rise of up to 30 K day−1 was diagnosed" -- this could be interpreted as a

heating rate (rate of change of temperature) or an ascent rate (expressed in terms of potential temperature). What is the intention -- to emphasise the large heating rate or the large ascent rate? If the latter then it might be clearer (particularly at this stage of the paper) to express it in terms of an equivalent rate of change of geometric heat.

It is a change of potential temperature, as shown in figure 4b of Khaykin et al. (2018). This is now mentioned in the text. We are somewhat reluctant to see a heating rate as a change of temperature, although it is a common and meaningful usage in many practical situation, as the first principle associates heating to a change of entropy, that is potential temperature.

l118: "This is true to a large extend" > "This is true to a large extent"

Correction done

l169: "This early stage is described in great details by Torres et al" > "detail" not "details".

Correction done